



# Influence of modes of climate variability on stratospheric gravity waves in the tropics using Radio Occultation and Reanalysis Data

Toyese Tunde Ayorinde[1], Cristiano Max Wrasse[1], Hisao Takahashi[1], Luiz Fernando Sapucci[2], Cosme Alexandre Oliveira Barros Figueiredo[1], Diego Barros[1], Ligia Alves da Silva[1], Patrick Essien[3], and Anderson Vestena Bilibio[1]

[1]Space Weather Division, National Institute for Space Research (INPE), São José dos Campos, SP, Brazil
[2]Instituto Nacional de Pesquisas Espacials (INPE), Centro de Previsão de Tempo e Estudos Climáticos, Rodovia Presidente Dutra, km 40, Cachoeira Paulista, SP,Brazil
[3]University of Cape Coast, Department of Physics, Meteorology and Atmospheric Research Lab, Cape Coast, Ghana

**Correspondence:** Toyese Tunde Ayorinde (toyese.ayorinde@inpe.br)

**Abstract.** The Intertropical Convergence Zone (ITCZ) is a critical driver of tropical climate, characterized by convective activity that transfers energy, influences atmospheric circulation, and modulates precipitation. These processes generate atmospheric disturbances, making the ITCZ a significant source of stratospheric gravity waves (GWs). This study investigates the relationship between the ITCZ and stratospheric GWs, as well as the influence of climate variability modes—Madden-Julian Oscillation (MJO), El Niño–Southern Oscillation (ENSO), and Quasi-Biennial Oscillation (QBO)—on GW activity and ITCZ dynamics. Using GNSS radio occultation (RO) data from COSMIC-1, COSMIC-2, and METOP satellites (2011–2021), we derive the latitudinal positions of the ITCZ and GW potential energy (Ep) maxima via Gaussian fitting. ERA5 and NCEP reanalysis data are used to validate ITCZ positions and estimate refractivity. Results show the ITCZ migrates $\sim10°$ latitudinally, with $\sim5°$ seasonal shifts between boreal winter and summer, while its strength remains relatively stable. Stratospheric GW Ep maxima exhibit seasonal patterns similar to the ITCZ, with smaller latitudinal gaps in the Northern and Southern Hemispheres. Multilinear regression reveals significant zonal variations in the impacts of QBO, ENSO, and MJO on ITCZ position and Ep, particularly over the American, African, and Asian sectors. ENSO and MJO drive substantial negative trends in ITCZ position, Ep, and refractivity, especially in Asian and African regions. However, zonal trends in Ep maxima and ITCZ positions remain stable, likely due to consistent Gaussian peak locations. Discrepancies in ITCZ trends and refractivity values between RO, ERA5, and NCEP data are attributed to differences in resolution, coverage, and assimilation techniques. This study highlights the complex interplay between the ITCZ, GWs, and climate variability modes.

## 1 Introduction

The tropical atmosphere is a complex system involving dynamic interactions between different atmospheric modes of climate variability that modify weather and climate on various temporal and spatial scales. These modes of climate variability include the Madden-Julian Oscillation (MJO), the El Niño-Southern Oscillation (ENSO), and the Quasi-Biennial Oscillation (QBO), which have a highly effective influence on the variability of the Intertropical Convergence Zone (ITCZ) (Münnich and Neelin,



2005; Kerns and Chen, 2018; Jin et al., 2023). QBO, ENSO, and MJO not only imply changes in the weather at the surface but also play an important role in the dynamics of the upper atmosphere, influencing the generation of convective gravity waves (GWs) in the equatorial stratosphere and their upward propagation. GWs play a vital role in improving the accuracy of climate model simulations, particularly in representing modes of climate variability. This is due to their interaction with sea surface temperature anomalies associated with key modes of variability, such as the Madden-Julian Oscillation (MJO), El Niño-Southern Oscillation (ENSO), and Quasi-Biennial Oscillation (QBO). A good representation of the QBO, for example, is essential for the quality of these simulations.

The ITCZ is a near-equatorial zone of convergence of the Northern and Southern Hemisphere trade winds characterized by convective activity and is an essential source of the GWs. Variability, location, and strength of the ITCZ are affected by the already mentioned modes of climate variability (Schneider et al., 2014; Kerns and Chen, 2018). The evidence of convective activity propagating along the intermediate axis of the eastern Pacific ITCZ has been studied using the characteristics of Kelvin waves, including its spatial structure propagation rate and characteristics of Kelvin waves (Straub and Kiladis, 2002). Dias and Pauluis (2009) studied the dynamics of GWs associated with convection propagating through a precipitation region analyzed in an idealized model of the large-scale atmospheric circulation. Several methods have been developed to determine the ITCZ (Zhang, 2001), including determining the location of maximum precipitation above a certain threshold, using visible and infrared satellite observations of highly reflective cloud cover, estimating wind field convergence, assessing relative vorticity structures, and using multiple variable criteria (Läderach and Raible, 2013). Läderach and Raible (2013) determined the ITCZ using specific humidity from two European Centre for Medium-Range Weather Forecasts (ECMWF) reanalyses dataset. In addition, Basha et al. (2015) determined the ITCZ using 13 years of refraction data from the Global Navigation Satellite System (GNSS) radio occultation (RO). Detecting the ITCZ is essential for various reasons (Zhang, 2001), including its role in the emergence of equatorial stratospheric GWs.

The equatorial stratospheric GWs are a significant component of the atmospheric system and a crucial driving mechanism in the lower and middle atmospheres through drag and diffusion processes. Deep convection and flow over topography are the two most common causes of equatorial stratospheric GWs (Alexander and Vincent, 2000; Alexander et al., 2000, 2008; Schmidt et al., 2008; Smith et al., 2020; Luo et al., 2021). Deep convective activities are the most likely source of equatorial stratospheric GWs in the Earth's tropical zone. Studies have linked deep convection processes to the origin of GWs. The ITCZ's increased convection, cloud formation, and rainfall lead to notable changes in heat flux, moisture, and momentum within and outside its boundaries, influencing tropical circulation. These physical characteristics are essential features in generating GWs in the equatorial region. While several studies have been conducted on GWs in the stratosphere, there is a lack of studies that specifically consider the link between atmospheric refractivity and equatorial stratospheric GWs. The latter plays a vital role in the upward transport of energy and momentum within the atmosphere, affecting processes like the stratosphere's circulation and the QBO. Understanding the relationship between atmospheric oscillations such as the QBO, MJO, ENSO, and stratospheric GWs is essential to understanding tropical atmospheric dynamics better. The ITCZ interacts with phenomena such as the circulation of the monsoon and the ENSO (Bain et al., 2011). There are several ways to identify the ITCZ region: the high precipitation, visible and infrared satellite observations, wind field convergence pattern, relative vorticity structures, etc.





The interactions of different modes of climate variability (MJO, ENSO, and QBO) with refractivity and GWs are critical in understanding global atmospheric dynamics. These modes of climate variability modulate the atmospheric circulation (including background wind and temperatures), influencing GW propagation across different air layers. The MJO, for example, affects tropical convection and can boost GW activity (Alexander et al., 2018). Depending on ENSO phases, the wave-mean flow interaction differs. El Nino is associated with more wave breaking in the lower stratosphere, strengthening circulation. In contrast, La Nina is associated with a reduced wave forcing, in particular convective gravity, due partly to the reduced convective activity and change in background wind (Konopka et al., 2016; Diallo et al., 2019). The alternating wind easterly and westerly QBO affect the vertical propagation of GWs, which are critical for generating stratospheric circulation patterns (Diallo et al., 2019, 2021). Understanding these interactions is essential in developing climate models and better constraining forecasting weather models since they contribute to the intricate linkage between the troposphere and stratosphere. The equatorial stratospheric GWs play significant roles in the vertical transport of energy and momentum, significantly affecting weather and climate. There are several studies conducted on the influence of atmospheric oscillations on GWs, for example, MJO (Moss et al. (2016); Alexander et al. (2018); Godoi et al. (2020); Wei et al. (2024)), ENSO (Geller et al. (2016); Liu et al. (2017); Godoi et al. (2020)), and QBO (Ern et al. (2014); Geller et al. (2016); Kang et al. (2020); Holt et al. (2022); Lee et al. (2024)), etc. Klotzbach et al. (2019) found a strong relationship between the QBO of equatorial stratospheric winds and the amplitude of the MJO during the boreal winter has recently been uncovered using observational data.

This study uses RO data, ERA5, and NCEP reanalysis data first to identify the ITCZ and the GW potential energy ($E_p$) maxima using the refractivity and temperature data, as well as to investigate the effects of modes of climate variability on stratospheric GWs within the ITCZ. The second goal is to understand how QBO, MJO, and ENSO modulate GWs within the ITCZ. Gravity wave studies require high-resolution RO vertical profiles of temperature, pressure, and humidity. Reanalysis products combine observations with model outputs, comprehensively describing atmospheric conditions over time. We organized the rest of this study as follows: The data subsection 2.1 describes the data. The methodology subsections 2.2 to 2.3 explains the methods for ITCZ identification and GW Ep. The results section 3 illustrates the seasonal changes in GW, its distribution, and possible links between equatorial stratospheric GW and the ITCZ. We also discuss in section 4 how ENSO, MJO, and QBO affect atmospheric refractivity and GW Ep. The conclusion in section 5 provide the final remarks.

## 2 Methodology

### 2.1 Data

The data used for this study is the reprocessed data set of dry temperature profiles from the satellite measurements of COSMIC -1, COSMIC -2, and METOP(a, b, c) (hereafter RO). The COSMIC-2 GNSS RO observation generates approximately 5000 to 7000 RO events daily, offering ample opportunities to analyze global atmospheric variations with high precision and accuracy. The COSMIC-2 mission generates various atmospheric parameters, such as the dry atmospheric temperature profile (atmPrf), wet atmospheric temperature profile (wetPrf) etc. The atmospheric profile "atmPrf" at level 2 of the COSMIC-2 data is processed operationally in near-real-time (nrt) without moisture information. Each atmPrf file includes high-resolution pro-





files of physical parameters like dry pressure, dry temperature, refractivity, bending angle, impact parameters, and geometric height above mean sea level. The RO data from RO were retrieved from the Data Analysis and Archive Center (CDAAC) website, which can be found at http://cdaac-www.cosmic.ucar.edu/cdaac/products.html. The RO data used for this study are from January 2011 and December 2021. The corresponding statistics of the available profile data used in this study are shown in Fig. 1.

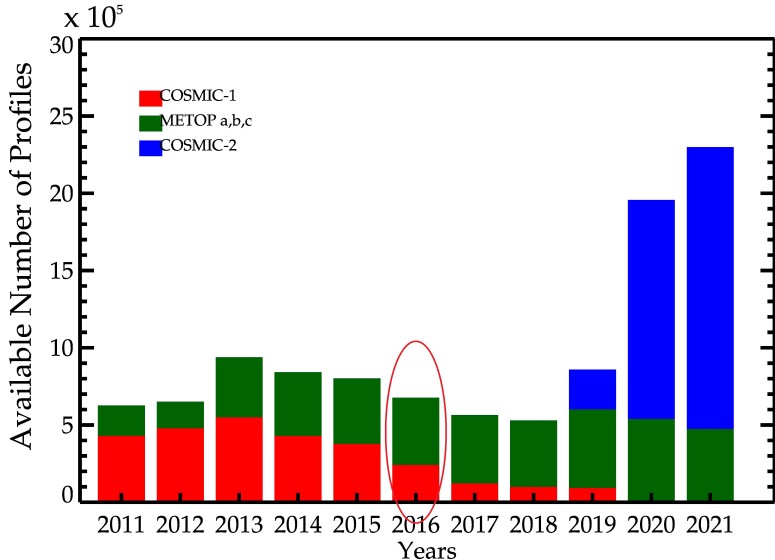

**Figure 1.** The yearly statistics of available COSMIC-1, COSMIC-2, the METOP temperature, and the refractivity profiles for 2011 and 2021. The COSMIC-1 profiles are red, all METOP profiles are green, and the COSMIC-2 profiles are blue. The randomly selected temperature and refractivity profiles (highlighted in the red oval) were used in that year to generate Fig. 5.

The total numbers of temperature ("atmPrf") and refractivity data ("wetPrf") profiles per year are presented in Fig. 1. The number of profiles per year exceeded 2 million in 2020 and 2021, attributed to the launch of COSMIC-2 in June 2019, resulting in a daily average of over 5000 profiles. Temperature and relative humidity were obtained from ECMWF Reanalysis v5 (ERA5) (https://www.ecmwf.int/en/forecasts/dataset/ecmwf-reanalysis-v5). The ERA5 is the most recent climate reanalysis conducted by the ECMWF. It integrates prior data and simulations to provide a consistent time series of numerous climatic variables.

ERA5 includes hourly data on various atmospheric, land-surface, and sea-state factors and uncertainty estimates. ERA5 uses standard latitude-longitude grids with a spatial resolution of $0.25° \times 0.25°$. The reanalysis contains data on air temperature, wind, rainfall, sea surface temperature, ocean wave height, and 37 pressure levels. ERA5 has been accessible since 1940 and continues to be extended ahead in time (Hersbach et al., 2020). Also, temperature and relative humidity were obtained from the National Centers for Environmental Prediction/National Center for Atmospheric Research (NCEP/NCAR) (https://psl.

noaa.gov/data/reanalysis/reanalysis.shtml) to estimate the atmospheric refractivity from a different source. The NCEP/NCAR Reanalysis is a comprehensive dataset that combines historical observations and numerical weather model simulations to provide reliable climate records. It gives 6-hourly and daily data from January 1948 to the present, including information





on several atmospheric characteristics such as air temperature, wind, and precipitation (Kalnay et al., 2018). The specific humidity, outgoing long-wave radiation (OLR), and vertical velocity data were obtained from the NOAA's (National Oceanic
and Atmospheric Administration) physical sciences laboratory website (https://psl.noaa.gov/data/gridded/data.ncep.reanalysis.html).

## 2.2 The atmospheric refractivity

The atmospheric refractivity ($N$) in a neutral atmosphere is a dimensionless quantity defined as $N = (n-1) \times 10^6$, where $n$ (unit less) is the atmospheric refractive index. $N$ is given by the following Eq. (1) and is primarily determined by water vapour
and temperature gradients in the lower atmosphere:

$$N = 77.6\frac{P_d}{T} + 72\frac{e_1}{T} + 3.75 \times 10^5 \frac{e_2}{T^2} \quad \text{(N-units)} ,$$ (1)

Where $P_d$ (hPa) is the dry atmospheric pressure, $e$ (hPa) is the water vapour pressure, and $T$ (K) is the absolute temperature. The first term of Eq. (1) is the dry term (Ndry) of the radio refractivity, and the second and third terms of this Equation are the wet terms (Nwet). Since dry and wet terms contribute to air refractivity, water vapour pressure (temperature) profiles require
distinct temperature (water vapour pressure) information. The data is obtained directly as one of the datasets from the "wetPrf" profiles of the RO data. The variations in mean refractivity obtained from 2011 to 2021 are presented in the result section. Also, N was estimated from the temperature and relative humidity at $850$ hPa using Eq. (1) from two reanalyses sources: the NCEP and EAR5 temperature and relative humidity data.

## 2.3 The GW potential energy

The atmospheric temperature profile ($T$) from RO is a function of height ($h$), which consists of the background temperature profile $\overline{T}(z)$ and the fluctuating component $T'(z)$. The $E_p$ has been used as a proxy for studying GW activities, as presented by Tsuda et al. (2000) and Ayorinde et al. (2023), and is given by:

$$E_p = \left(\frac{g}{N}\right)^2 \left(\frac{T'}{\overline{T}}\right)^2 ,$$ (2)

Where $g$ is the acceleration due to gravity, $N$ is the Brunt-Väisälä frequency, and $\bar{T}$, and $T'$ are the local background tempera-
ture profiles and the temperature fluctuation profiles caused by GW activities, respectively. The $E_p$ calculation is based on the accurate extraction of $T'$, which is given by:

$$T' = T - \overline{T},$$ (3)

And $N$ is given as follows:

$$N^2 = \frac{g}{\overline{T}}\left[\frac{\partial \overline{T}}{\partial h} + \frac{g}{C_p}\right],$$ (4)



$C_p$ is the specific heat capacity of air at constant pressure, and $h$ is the altitude. The GW $\mathrm{E_p}$ is calculated using Eq. (2). The $\mathrm{E_p}$
depends only on the raw temperature profile, which can be separated into the background temperature ($\overline{T}$) and the temperature
fluctuation ($T'$). In calculating $\mathrm{E_p}$, the $T'$ is the main issue that requires careful attention (Ayorinde et al., 2023). The raw
temperature profile obtained from RO is first interpolated at $100\ \mathrm{m}$ intervals along the altitudes. Each temperature profile is
divided into cell sizes of $20° \times 10°$ of longitude and latitude between 10 and $50\ \mathrm{km}$ of altitude, and the mean temperature of

each grid is calculated. The mean temperature profile is decomposed using a continuous wavelet transform (CWT) (Torrence
and Compo, 1998; Moss et al., 2016) to obtain the background temperature ($\overline{T}$). The $\overline{T}$ is interpolated back to the positions of
temperature profiles and subtracted from the raw temperature profile using Eq. (3) to obtain $T'$. The variations of the mean $\mathrm{E_p}$
obtained from 2011 to 2021 are presented in Fig. 4.

## 2.4   Method of identifying ITCZ and maximum Ep

The water vapour contributes to greater atmospheric refractivity ($N$) than the atmospheric temperature in the upper troposphere
region approximately 8-12 km. Nonetheless, the RO satellite's refractivity data is precise from the surface to $40\ \mathrm{km}$. It also
contains temperature and water vapour data (Basha and Ratnam, 2009; Ratnam and Basha, 2010). Ratnam and Basha (2010)
and Basha et al. (2015) used RO refractivity to locate the ITCZ. It is worth noting that changes in the lower troposphere
can considerably impact refractivity. In contrast, temperature regulates refractivity in the higher troposphere due to less water

vapour input. As a result, it is feasible to detect the ITCZ from the surface to the high troposphere using refractivity, a unique
metric available only directly from RO satellites.

In this study, we employed a similar procedure used by Läderach and Raible (2013) to locate the ITCZ position concerning
the resolution of the satellite data by fitting Gaussian functions in our methodology. We also used it to determine the position
of the maximum GW $\mathrm{Ep}$. This approach determines the meridional distribution of refractivity and $\mathrm{Ep}$. The peak position,

location, and width (full-width half maximum) of the Gaussian are computed using the refractivity from RO, NCEP, and ERA5
data and Ep data grids for each longitude.

The data series includes global refractivity and Ep observations with $2° \times 2°$ longitude and latitude grid resolution. We
eliminate the irregular border layers by calculating the mean of the refractive index and Ep at each $10°$ latitude along the
longitudes to reduce the effects of orography. This approach was applied to the monthly averages of the refractive index and

Ep at a pressure level of $850\ \mathrm{hPa}$. The parameters of the refractive index distribution are determined by fitting a Gaussian
function at each longitude, provided as:

$$Q(\phi) = Q_{\max} e^{-\left\{ \dfrac{(\phi - \phi_{\max})^2}{2\sigma_Q^2} \right\}}, \tag{5}$$

where $Q(\phi)$ represents the meridional refractive index at each longitude; $\phi_{\max}$, represents the meridional position of the
particular refractive index at a given longitude; $Q_{\max}$ represents the highest value of refractive index, and $\sigma_Q^2$ represents the

meridional variance of refractive index.



## 2.5 Trend analysis method

The linear trend of Ep and refractivity zonal mean time series and the responses to ENSO, MJO, and QBO wind are calculated for each longitude using the multivariate linear regression (MLR) approach (Wolter and Timlin, 2011; Li et al., 2013; Ayorinde et al., 2024). The MLR can determine the link between one dependent variable (e.g., Ep) and two or more independent variables
(e.g., ENSO, MJO, and QBO wind). The MLR method's equation is as follows:

$$
\begin{aligned}
\Psi(t_{i,j}) = & \mu + \alpha_0 \dot{t}_{i,j} + \alpha_1 \cdot QBO_{30mb}(t_{i,j}) + \alpha_2 \cdot QBO_{50mb}(t_{i,j}) + \alpha_3 \cdot (t_{i,j}) + \alpha_4 \cdot ENSO(t_{i,j}) \\
& + \alpha_5 \cdot RMM(t_{i,j}) + \text{Residual},
\end{aligned}
\tag{6}
$$

With $\quad i = 2011, 2012, \ldots, 2021; \quad$ and, $\quad j = 1, 2, \ldots, 12$

The monthly zonal mean value of Ep or refractivity is represented as $\Psi$, the monthly values denoted as $(t_{i,j})$ (where $(i)$ is the year and $(j)$ is the month), and the quantity $\mu$ represents a constant Ep value. The parameter $\alpha_0$, which depicts the change in GW parameters over time, represents the monthly zonal Ep or refractivity linear trend from 2011 to 2021. The parameters
$\alpha_1$, $\alpha_2$, $\alpha_3$, and $\alpha_4$ show the relationship between the time series of the Ep or the refractivity parameters and the time series of the four indices, depicting the monthly GW parameter's zonal response to QBO at 30 hPa and 50 hPa, MJO, and ENSO indices. The residual of the regression model can be utilized to estimate each coefficient's standard deviation and p-value. This estimation can be achieved using the variance-covariance matrix and the Student t-test (Kutner et al., 2004; Mitchell et al., 2015). Fig. 2 presents the reference time series of ENSO, MJO, and QBO from 2011 to 2021. The bi-monthly Multivariate
ENSO Index (MEI) values in Fig. 2a show the temporal variations of the ENSO phases (Wolter and Timlin, 2011). The monthly Real-time Multivariate MJO Index (RMM) values in Fig. 2b show the temporal variations of the RMM phases. The RMM is calculated by projecting 20-96 day filtered OLR, which includes all eastward and westward wave numbers, onto daily spatial EOF patterns of 30-96 day eastward filtered OLR (Hoffmann et al., 2021). In Fig. 2c, the reference time series for 30 hPa and 50 hPa are the temporal fluctuation of QBO zonal-mean zonal wind across the equator (Gavrilov et al., 2002). The QBO
data are from radiosondes (obtained from the Meteorological Service Singapore Upper Air Observatory (1.34°N, 103.89°E), NASA (National Aeronautics and Space Administration)/GMAO (Global Modeling and Assimilation Office) assimilated data, and NASA satellites through the NASA earth-data site.



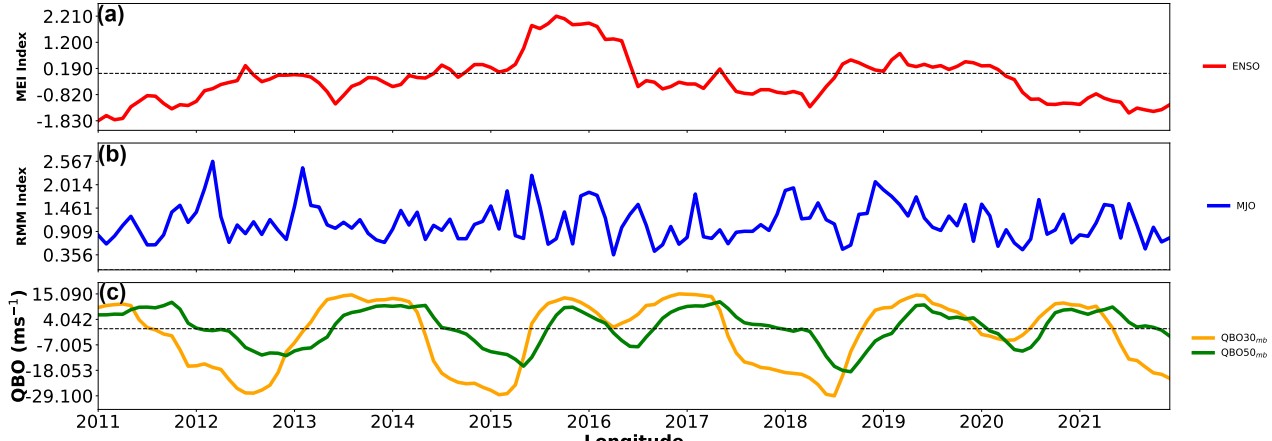

**Figure 2.** Reference time series from year 2011 to 2021 used for the regression analyses. **(a)** Multivariate ENSO Index (MEI) to characterize the ENSO signal (red), **(b)** The amplitude of Real-time Multivariate MJO indices (RMM) 1 and 2 to characterize the MJO signal (blue), and **(c)** 30 hPa (24 km, orange) and 50 hPa (21 km, green) zonal winds over the equator to characterize the QBO.

## 3   Results

The global distribution of refractivity values from RO, ERA5, and NCEP at 850 hPa, integrated over a $2° \times 2°$ latitude-
longitude grid for the year 2021, is shown in Fig. 3. In the paper, we divided the seasons into December–January–February (DJF), March-April-May (MAM), June–July–August (JJA), and September-October-November (SON). The analysis is presented for boreal winter (DJF) and summer (JJA). We observed that refractivity ($N$) varied between 260 and 290 N units (Fig. 3a). In contrast, refractivity from ERA5 and NCEP ranged from 240 to 310 N units (Fig. 3a and b, respectively), capturing the Intertropical Convergence Zone (ITCZ) around the equatorial region. At the 850 hPa pressure level, cloud-top heights
are inferred from N profiles. Convective regions are prominent across the Indian Ocean, the western Pacific, Africa, and South America during DJF. The higher refractivity values shift to the Southern Hemisphere (SH) in DJF (Fig. 3a) and to the Northern Hemisphere (NH) in JJA (Fig. 3b). Key regions with relatively high refractivity (highlighted with red circles) include South America and Africa during DJF, as well as Mexico, the southern United States, and the Asian monsoon region during JJA. The refractivity values from RO, ERA5, and NCEP exhibit similarities, although ERA5 and NCEP tend to show slightly higher
values, indicating an overestimation of refractivity by approximately two tens of N units compared to RO.



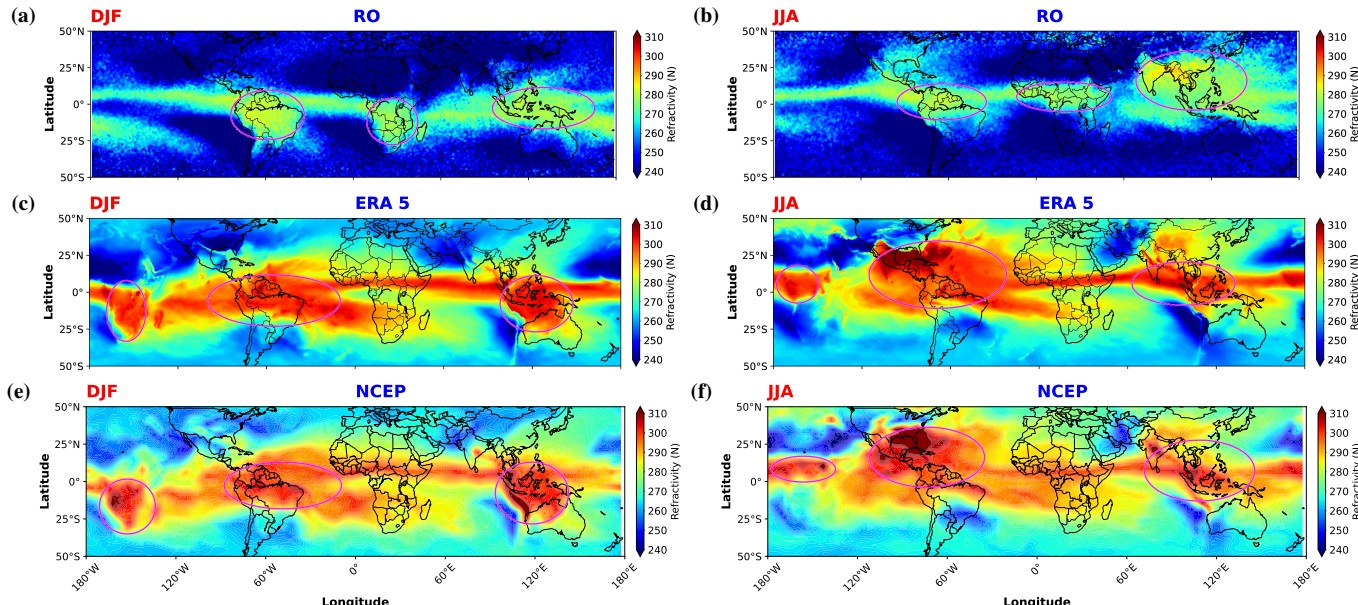

**Figure 3.** The refractivity in 2021 during December-January-February (DJF) (**a, c, e**) and June-July-August (JJA) (**b, d, f**) at 850 hPa. The circled regions are the regions of high concentrations of refractivity. (**a** and **b**) The refractivity was obtained from RO for DJF and JJA. (**c** and **d**) The refractivity is estimated from ERA5 for DJF and JJA. (**e** and **f**) The refractivity estimated from NCEP for DJF and JJA.

The global distribution of Ep values measured at 20 km in DJF and JJA in 2021 (a year with the highest number of profiles) is given in Fig. 4. The Ep contour intervals are 0 - 15 $\mathrm{J\,kg^{-1}}$ at 20 km. Higher values of ~15 $\mathrm{J\,kg^{-1}}$ at 20 km for both seasons are seen towards the equator and decrease towards the polar regions. The high values in the equatorial region (Fig. 4) are attributable to the significant quantity of deep convective activity caused by high temperatures and humidity in that region.

In Fig. 4, Ep values at tropical and subtropical latitudes are substantial in all the months presented. This might be related to more convection than predicted and partially due to equatorial waves. The GW Ep is considerably high in JJA over the southern Andes (30°S-50°S) and diminishes eastward in JJA Fig. 4b. This occurrence might be owing to the eastward spread of orographic mountain waves created by the Andes' north-south distribution, primarily due to low-level westerlies and the persistent jet (Alexander et al., 2010). We found a similarity between the Asian monsoon and the eastern Pacific Ocean in the

refractivity in Fig. 3 and the high values of the GW Ep in Fig. 4. The equatorial stratospheric GW activity is centred towards the equator over Southern America and Africa (Fig. 4), compared to the result in Fig. 3. Moderate values of Ep ($\sim$ 6-7 $\mathrm{J\,kg^{-1}}$) showing equatorial stratospheric GW activity are seen over Mexico and the USA with high refractivity values in JJA.





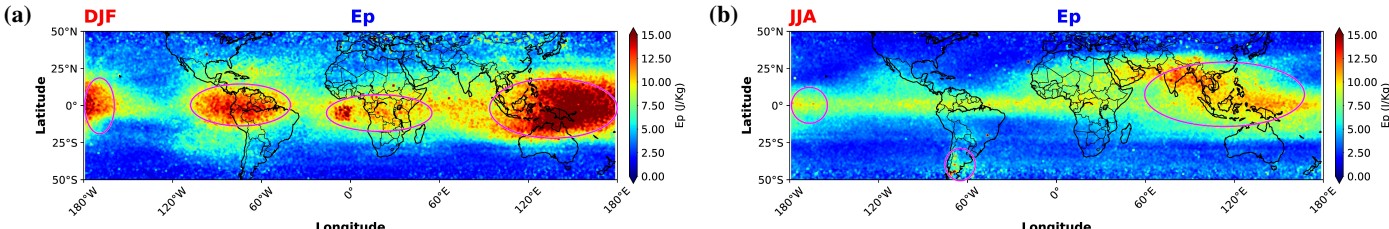

**Figure 4.** The Ep variations in 2021 during December-January-February (DJF) **(a)** and June-July-August (JJA) **(b)** at 20 km. The circled regions are the regions of high concentrations of Ep.

## 3.1 The ITCZ and Ep maxima

Based on the approach used by Läderach and Raible (2013); Basha et al. (2015) to identifying ITCZ and maximum Ep, we

report the morphology, inter-annual variability, and possible trends of the refractivity-derived ITCZ and the local maxima of the equatorial stratospheric GW Ep distribution. The emphasis is on DJF and JJA. For example, the technique based on monthly mean Ep and refractivity at 850 hPa is validated over DJF and JJA. The weighted zonal distribution of Ep and refractivity at 76°W in 2016 (location and years were chosen at random) and the resulting Gaussian fit for DJF and JJA are shown in Fig. 5. The distribution of Ep and refractivity is similar, with the highest near the equator during JJA and a movement towards the

north (~15°N for both Ep and refractivity) in JJA. Nonetheless, there is a slight variation in the northern latitudes in JJA. The mean maximum location was determined by refractivity in 2016 throughout the DJF and JJA seasons, as shown in Fig. 5. During DJF, the maximum location changes to the SH to about 7°S for both Ep and the refractivity in 2016 at 76°W.





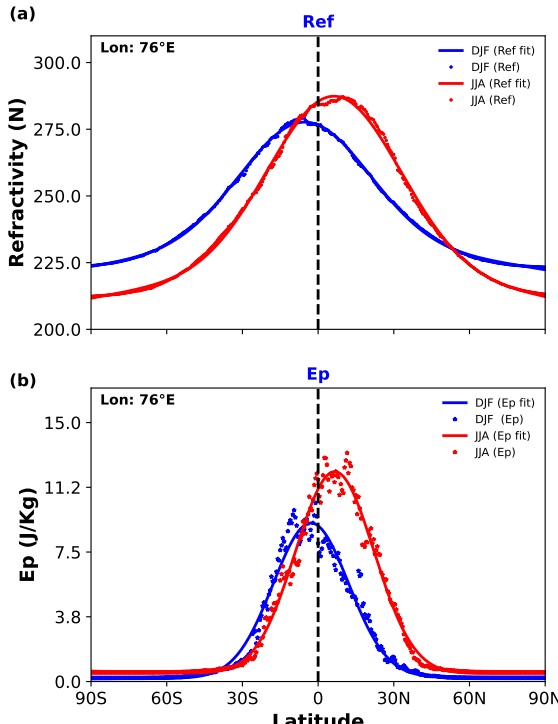

**Figure 5.** Latitudinal distribution of **(a)** refractivity and **(b)** Ep observed during December- January-February (DJF) and June-July-August (JJA) at $76°$W in 2016 at $800$hPa. Thick lines show the Gaussian fit.

The ITCZ's mean shape and inter-annual seasonal variations, as determined by refractivity data and the seasonal maximum Ep, are presented in Fig. 6. The inter-annual variability of ITCZ using RO, ERA5, and NCEP data was computed across DJF

and JJA months from 2011 to 2021 (11 years of observation). Inter-annual variability has shifted by nearly $\sim 5°$ - $15°$ north and south. Throughout the 11 years of data, the ITCZ positions are practically similar. There is a slight difference in the ITCZs obtained from RO, ERA5, and NCEP. From 2011 to 2021, Fig. 6b depicts the mean positions of maximum refractivity and Ep for DJF and JJA. Throughout DJF and JJA in both NH and SH, the mean distribution of ITCZ exhibits a slightly noticeable movement. The ITCZ is positioned about $5°$S in DJF and approximately $7°$N in JJA. The ITCZ position moves by $10°$ between

DJF and JJA throughout the global oceanic areas. The SON and MAM are the transition seasons between DJF and JJA, so their results are expected to be similar. Apart from seasonal oscillations, the ITCZ has inter-annual shifts in location and intensity.



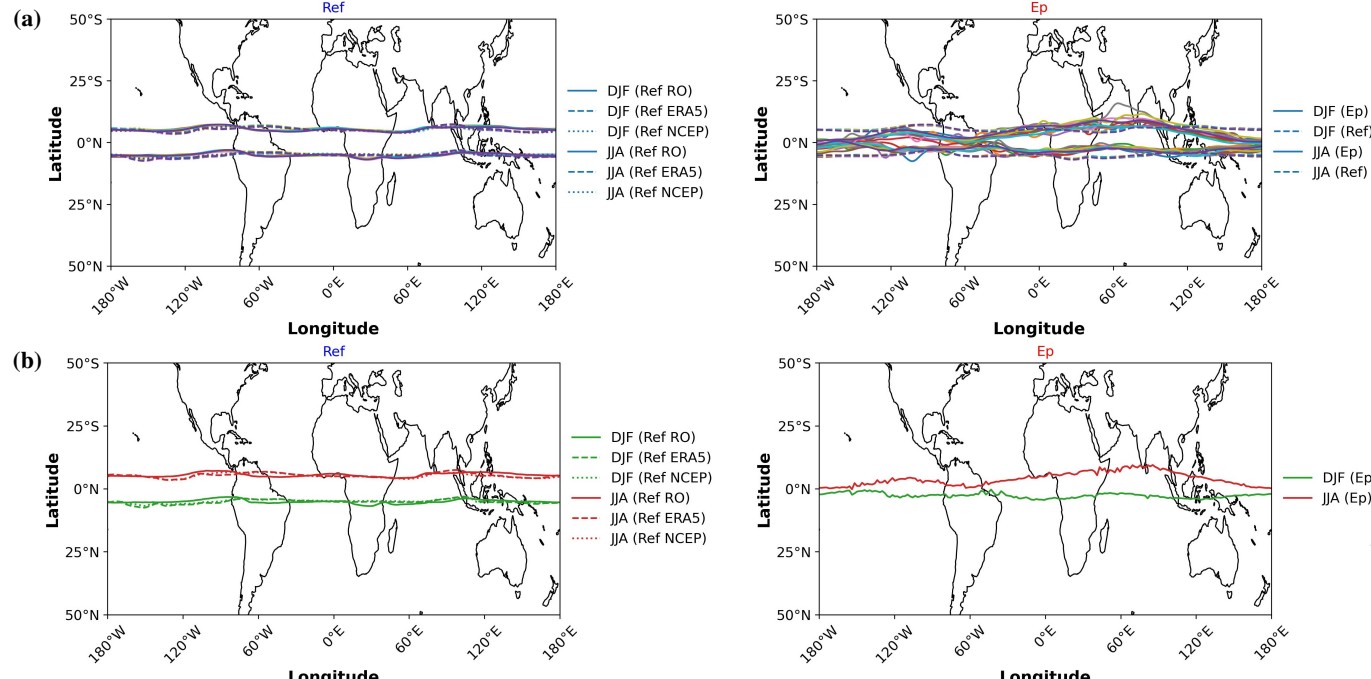

**Figure 6.** ITCZ and Ep maxima are located from 11 years of refractivity data and Ep maxima of the equatorial stratospheric GW at 20 km. **(a)** The global inter-annual variability of ITCZ lines during JJA and DJF of the refractivity and Ep maxima. **(b)** The estimated mean of the ITCZ and Ep maxima lines during JJA and DJF.

The global equatorial stratospheric GW Ep maxima also showed the same seasonal differences as the ITCZ. The global equatorial stratospheric GW Ep maxima (Fig. 6b) showed more consistency in the NH and SH with much closer spacing than the ITCZ. We find two essential features that distinguish the Ep morphology from the ITCZ. The first one is related to the shift

of Ep position by $5°$ between DJF and JJA. The distance between the Ep maxima is consistent, with JJA moving northward and DJF southward. Secondly, the Ep maxima in DJF and JJA showed two points of convergence over the South American Amazon and the equatorial Pacific. These convergences do not exist in MAM or SON. The South American Amazon and Central American climate are the broadest deep convective zones around the globe (Garcia and Kayano, 2010; Boers et al., 2013; Rojas et al., 2016; García-Franco et al., 2020), and these could be the reason for the convergence of Ep maxima. It is

also noted that the convergence over the Pacific Ocean could result from the double ITCZ prevailing in this region (Hwang and Frierson, 2013; Zhang et al., 2019; Tian and Dong, 2020). This feature is also seen in Fig. 3 over the Pacific Ocean. The most significant gap in seasonal localization between the ITCZ in DJF and JJA is seen over Africa and the Indian Ocean to the east, also observed in Ep maxima. This result could be due to the Sun being directly overhead at the Tropic of Capricorn during DJF in the Southern Hemisphere (SH), leading to a shift of the Intertropical Convergence Zone (ITCZ) and the convergence

of trade winds further into the SH. This results in the formation of deep convective cloud systems. During this period, a low-





pressure area develops south of the equator, while a high-pressure area forms north of the equator, driving the ITCZ southward due to clear-sky forcing. Conversely, when the Sun is directly overhead at the Tropic of Cancer during JJA, the ITCZ shifts northward, concentrating trade winds in the Northern Hemisphere (NH), which could also influence the generation of equatorial stratospheric GWs.

## 3.2 Equatorial stratospheric GWs, ITCZ and other related parameters

In this section, we compare the two main parameters (i.e., specific humidity and vertical wind) in different seasons, which should show whether the locations of the derived ITCZ and Ep maxima match. The first parameter is the mean specific humidity, overlaid in Fig. 7a and b with the ITCZ, Ep maxima, and OLR at $200 - 250 \, \mathrm{W\,m^2}$. The specific humidity, the vertical wind velocity, and the OLR are taken at $800 \, \mathrm{hPa}$. In the lower troposphere, atmospheric humidity is an important parameter.
The amount of water vapour per air volume determines the humidity, which is determined by evaporation, advection, and precipitation. Our results show perfect agreement between OLR and maximum specific humidity, especially in DJF. There are exceptions in JJA, particularly over the southern United States of America (USA), where specific humidity is high outside the OLR, despite the region experiencing opposite seasons. The OLR shows that the ITCZ is a wandering, thin band of clouds around the equator.

As previously reported, Läderach and Raible (2013) used specific humidity to determine the position of the global ITCZ, which was later confirmed by Basha et al. (2015). For example, Holloway and Neelin (2009) showed a positive relationship between lower tropospheric humidity and tropical deep convection, the latter associated with radiative heating (cooling) at the surface (top). Bengtsson (2010) consistently found extensive evaporation on both sides of the ITCZ that carries water vapour into the ITCZ, a first indication of the link between tropical deep convection in the ITCZ and moisture in the lower troposphere.
We also observed divergence of the ITCZ from the peak specific humidity values over the African landscape and the Indian Ocean. This phenomenon will be discussed in the discussion section. The closest match between the ITCZ and Ep maxima was observed over the Asian monsoon.

Another approach to identifying the ITCZ is wind convergence, later associated with the vertical wind by Žagar et al. (2011). The vertical wind indicates the ITCZ position, which is identical to the OLR position. Fig. 7c and d shows the 2D variation of
the vertical wind superimposed on the ITCZ and Ep maxima in all seasons. The black contour lines show the vertical velocity at $0 \, \mathrm{m\,s^{-1}}$, which can also be used as a proxy for the ITCZ (Basha et al., 2015). The zero vertical wind can be considered the convergence of the wind. In all seasons, the ITCZ is mainly within the wind convergence. As mentioned earlier, there is a slight variation between the ITCZ and the wind convergence over Africa and the Indian Ocean. Interestingly, the Ep maximum fits perfectly within the zero vertical wind. During the JJA, the Ep maxima line shifts southward over South America, further from
the ITCZ and wind convergence. This phenomenon could be a consequence of ENSO or other tropical dynamics.





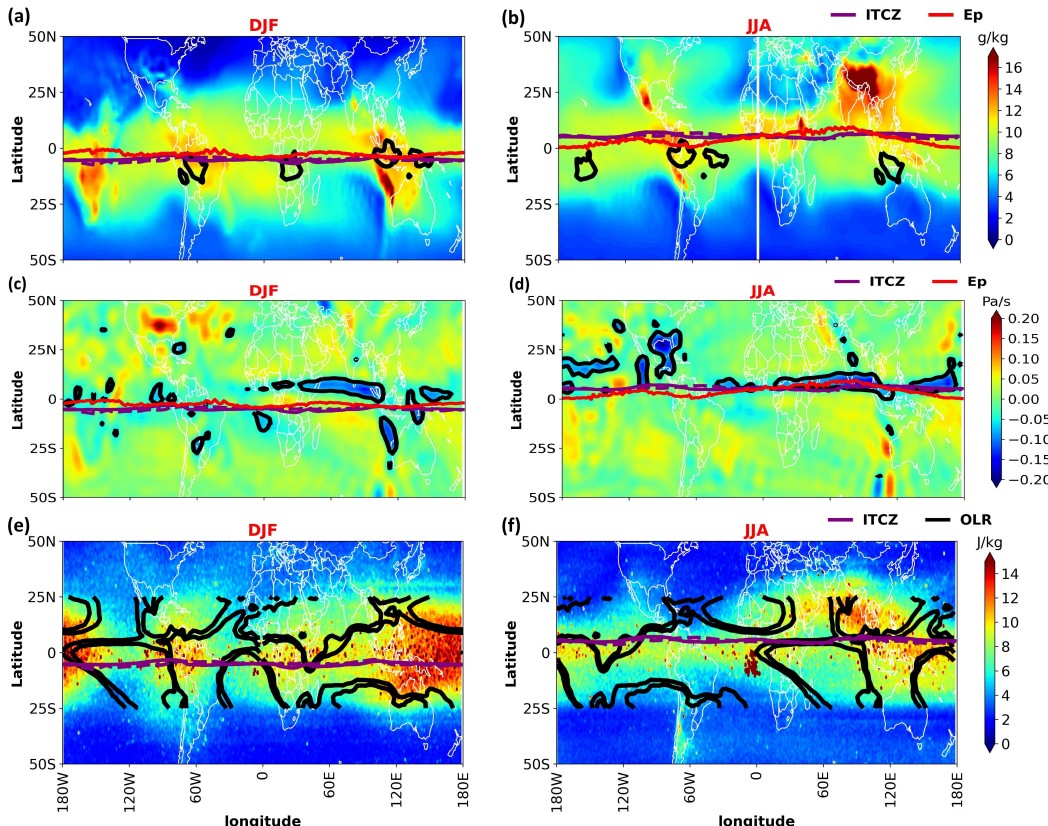

**Figure 7.** The mean global specific humidity (in $\mathrm{g\,kg^{-1}}$), vertical wind velocity (in $\mathrm{Pa\,s^{-1}}$), and Ep (in $\mathrm{J\,kg^{-1}}$) from 2011 to 2021 in December-January-February (DJF) and June-July-August (JJA). (**a** and **b**), The mean global specific humidity was overlaid with the Ep maxima (red solid line) and ITCZ (purple solid line) during DJF and JJA, respectively. The black contour lines are the OLR at 200 - 250 $\mathrm{W\,m^{-2}}$. (**c** and **d**), The mean global vertical velocity was overlaid with the Ep maxima (red solid line) and ITCZ (purple solid line) (top row) during DJF and JJA, respectively. The black contour lines are the vertical velocity at $0\,\mathrm{m\,s^{-1}}$ (middle row). (**e** and **f**), The mean global Ep was Ep overlaid with the ITCZ (purple solid line) during DJF and JJA, respectively. The black contour lines are the OLR at 200 - 250 $\mathrm{W\,m^{-2}}$. The specific humidity, the vertical wind velocity, and the OLR are taken at 800 hPa.

To verify that the location of the ITCZ derived from the refractivity field and OLR was consistent, we plotted the refractivity field and Ep maxima on the mean Ep distribution shown in Fig. 7e and f. Our result showed that the ITCZ derived from the refractivity field coincides with the equatorial OLR at 200 - 250 $\mathrm{W\,m^{-2}}$. It should be noted that OLR is electromagnetic radiation with wavelengths in the range of 3 - 100 μm radiated from the Earth and its atmosphere into space as thermal radiation. The OLR represents the atmosphere's total radiation, making it vital to Earth's energy budget (Chakraborty et al., 2018). The equatorial OLR is a proxy for estimating the global deep convective zones, sometimes called the ITCZ (Oke, 2002; Pincus, 2004). The mean Ep fluctuation has the same features described in Fig. 4 above. In Fig. 7e and f, the equatorial stratospheric






GW Ep in DJF is mainly concentrated in four regions globally. Ep values are high regardless of land and sea proximity to deep convection areas. Deep convection can be detected in Africa, South America, Asia, and the Indian and Pacific Oceans.

Our results show a one-to-one correlation between equatorial stratospheric GW activity and deep convection in the band of latitudes where deep convection is detected. This result agrees with the results of Baldock and Huntley (2002); Ratnam et al. (2004).

In JJA (Fig. 7e and f), equatorial stratospheric GW is mainly concentrated in the eastern part of Africa, Asia, and the Indian and Pacific Oceans. equatorial stratospheric GW activity has shifted toward the NH, concentrating mainly on India and

Southeast Asia, then dropping toward the Pacific. This phenomenon is seen as a result of the ITCZ and trade winds in this region, as reported earlier. The equatorial stratospheric GW decreased mainly over western and equatorial South America and did not precisely follow the ITCZ path as observed in DJF. This result could result from ENSO producing more dryness over tropical South America in JJA (Gavrilov et al., 2004). This phenomenon produces a narrower band of equatorial stratospheric GW activity concentrated along a line in the NH closer to the equator. Ratnam et al. (2004) reported that anomalies of Ep and

OLR from their zonal averages imply that negative (positive) values of Ep and OLR reflect a reduction (increase) or increase (decrease) of GW energy and convection, respectively, and discovered significant relationships between GW activity and OLR measurements (Byrne et al., 2018).

### 3.3   Modes of climate variability and ITCZ and Ep trends

In this section, we presented the zonal trends of the 11-year ITCZ derived from RO, ERA5, and NCEP refractivity and the

GWs Ep in Fig. 8a and b, also the zonal correlation coefficients between the ITCZ and the Ep in Fig. 8c. Fig. 9 presented the relationships between the various atmospheric oscillations (ENSO, MJO, and QBO) and the ITCZ derived from RO, ERA5, NCEP refractivity, and Ep maxima latitudinal locations. Fig. 10 depicts the zonal refractivity and Ep values at the Gaussian peak. We also presented in Fig. 11 the zonal trends refractivity from RO, ERA5, and NCEP and the Ep values at their Gaussian peak in Fig. 10a and b, and the zonal correlation coefficients between the refractivity and the Ep values in Fig. 10b. Fig. 12

presented the relationships between the atmospheric oscillations and the refractivity from RO, ERA5, NCEP, and Ep values.

#### 3.3.1   *ITCZ and Ep maxima location Trends*

The trends of the ITCZ derived from RO, ERA5, and NCEP refractivity and maximum Ep latitudinal locations are shown in Fig. 8. The ITCZ in Fig. 8a showed different positive trends ($\sim$0.005-0.011 N/month) at different zonal locations. The trends exhibit variation across the longitudinal sectors. The American sector ($\sim$120°W to 60°W) displays generally constant

patterns throughout the datasets. Still, the African sector ($\sim$60°W to 60°E) diverges, with a significant drop in the trend for ERA5. Asian sector ($\sim$60°E to 120°E) trends are consistent across datasets, with a modest rise reported. Unlike ITCZ trends, the maximum Ep position exhibits a unique oscillation pattern over longitudes Fig. 8b. The trends are typically positive in the African ($\sim$0.006-0.014 J kg$^{-1}$ per month) and Asian sectors ($\sim$0.001-0.009 J kg$^{-1}$ per month) but turn negative in the American sector ($\sim$−0.004 J kg$^{-1}$ per month) around 60°W and at eastern Pacific. The negative trend in the Ep over the

stratosphere in the American sector has earlier been reported by Ayorinde et al. (2024). The transitions between positive and





negative trends demonstrate how Ep latitudinal movements may be affected by regional differences. The zonal coefficients that relate the ITCZ, as determined by RO, ERA5, and NCEP refractivity, with Ep maxima latitudinal locations show how strongly the ITCZ corresponds with shifts in Ep locations (Fig. 8c). The coefficients are often larger in the African and Asian sectors, indicating a stronger link between ITCZ and Ep in these regions than in the American sector. The zonal correlation

between ITCZ and the GW energy is generally positive, with a correlation above 60% over the Asian monsoon areas. As expected, it showed that most GW activity is generated by mesoscale convective systems in this region (Liu et al., 2017). The low correlations between eastern South America and the western Pacific are a subject of interest, and they could result from associated atmospheric oscillation that influences both ITCZ and the GWs.

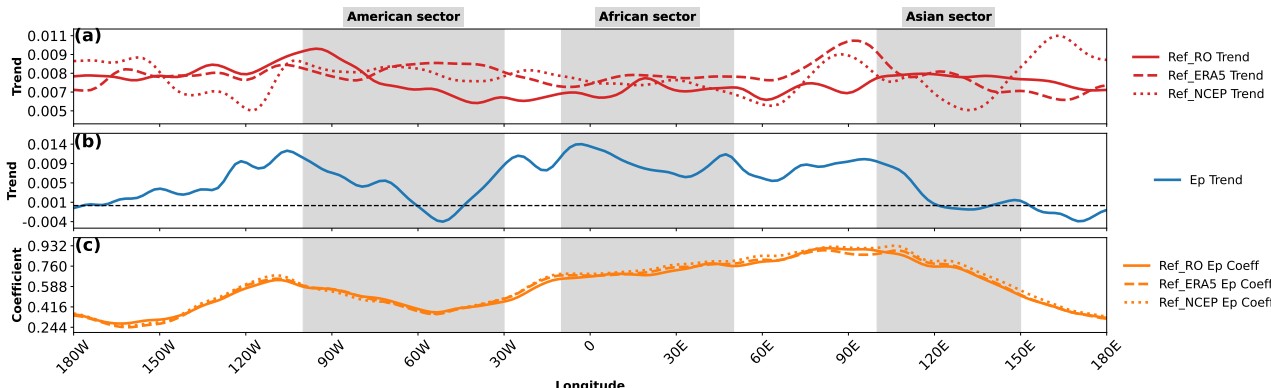

**Figure 8.** The zonal trends and ITCZ coefficients derived from RO, ERA5, NCEP refractivity, and maximum Ep latitudinal locations over 11 years. **(a)** The zonal trends ITCZ derived from RO, ERA5, and NCEP refractivity. **(b)** The zonal trends of the maximum Ep latitudinal location. **(c)** The zonal coefficients of the ITCZ derived from RO, ERA5, NCEP refractivity, and maximum Ep latitudinal locations. The shaded areas mark the American (100°W-30°W), African (10°W-50°E), and Asian sector (100°E-150°E) sectors are marked with shaded regions. The black horizontal dotted lines are the zero demarcation lines.

### 3.3.2 *Modes of climate variability and ITCZ and Ep maxima locations*

Fig. 9a depicts zonal regressions of the maximum GW Ep latitudinal locations that include three atmospheric oscillations: ENSO, MJO, and QBO at different pressure levels (30 and 50 mbar, respectively). The regression coefficients between the maximum Ep latitudinal position and the ENSO coefficients shown in Fig. 9a vary longitudinally, with positive values (0.7 J kg$^{-1}$ per month) in the Asian sectors but negative values ($-0.3$ J kg$^{-1}$ per month) in the American sector and near zero in the African sector. The maximum Ep latitudinal position advances northward during positive ENSO phases (El Niño) in the

Asian sectors but somewhat southward in America and Africa. The variation across longitudes indicates regional variances in how ENSO affects GW activity (Geller et al., 2016; Rakhman et al., 2017; Kawatani et al., 2019). The regression coefficients between the highest Ep latitudinal location and the MJO are shown in the Fig. 9b. The MJO coefficients are consistently negative ($\sim -0.5$-$-3$ J kg$^{-1}$ per month) across all sectors, with larger amplitude in African and Asian sectors. This shows





that during active MJO periods, the maximum Ep latitudinal location shifts southward, with a more significant effect in the
Eastern Hemisphere. The MJO's consistent negative trend suggests that it has a more uniform influence on GW activity across
longitudes than ENSO. The regression coefficients for the most significant Ep latitudinal location and the QBO at 30 and 50
mbar are shown in Fig. 9a. The coefficients at both pressure levels exhibit similar longitudinal trends ($\sim -0.095$-$0.008$ J kg$^{-1}$
per month) but differ in magnitude. In the American sector, the QBO coefficients are around zero or slightly positive ($\sim 0.008$
J kg$^{-1}$ per month), suggesting minimal influence. However, in the African and Asian sectors, the coefficients turn negative,
suggesting a southward shift in the maximum Ep latitudinal location during the QBO's westerly phase. The higher values at
50 mbar indicate that the lower stratosphere has a more considerable effect on GW activity than the upper stratosphere (30
mbar). The findings imply that ENSO, MJO, and QBO all play different roles in the zonal variability of localized region GW
Ep concentrations, with each oscillation having a more significant impact on specific sectors.

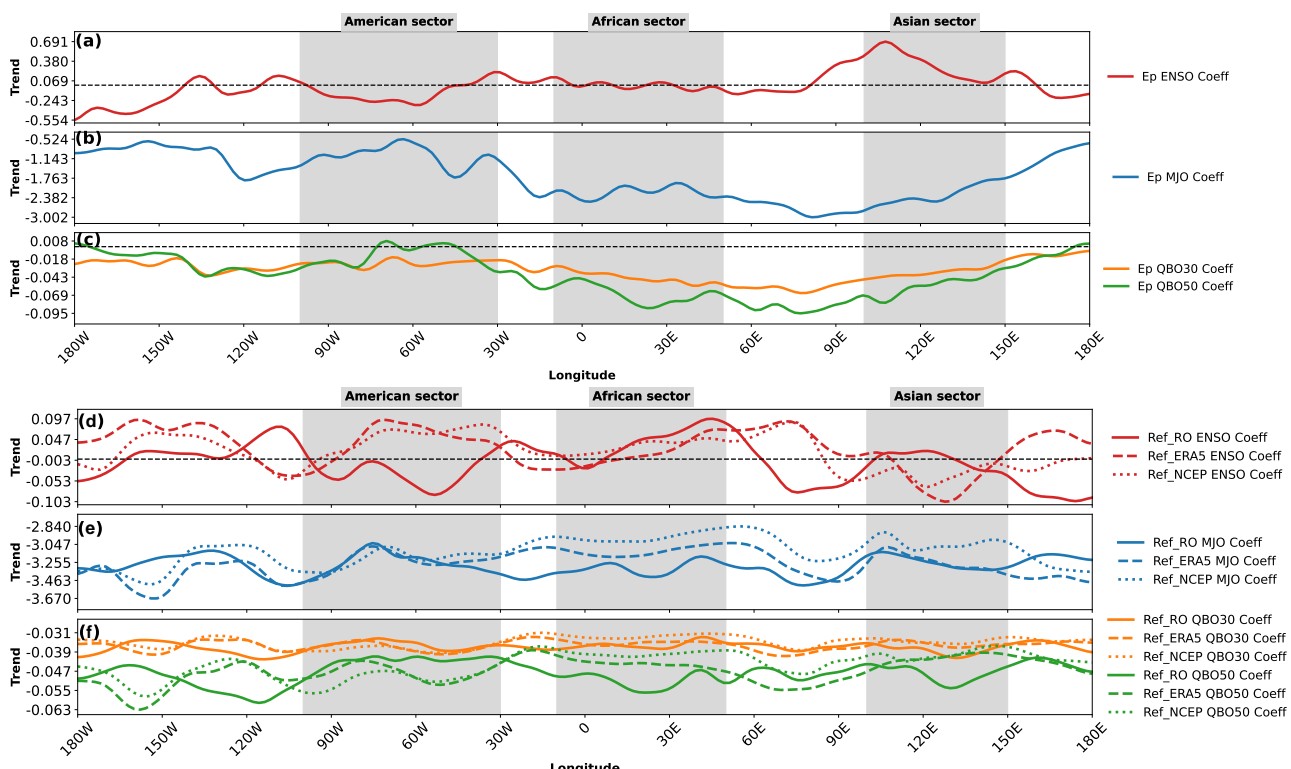

**Figure 9.** The zonal regressions of the atmospheric oscillations with the ITCZ derived from RO, ERA5, NCEP refractivity, and maximum
Ep latitudinal locations over 11 years. **(a-c)** The zonal regressions of the maximum Ep latitudinal locations with ENSO (**a**, in red lines), MJO
(**b**, in blue lines), and QBO at 30 mbar (**c**, orange lines) and at 50mbar (**c**, green lines). **(d-f)** The zonal ITCZ derived from refractivity
measurements by RO (solid lines), ERA5 (dashed lines), and NCEP (dotted lines) for ENSO (**d**, in red lines), MJO (**e**, in blue lines), and
QBO at 30 mbar (**f**, orange lines) and 50 mbar (**f**, green lines). The shaded areas mark the American ($100°$W-$30°$W), African ($10°$W-$50°$E),
and Asian sector ($100°$E-$150°$E) sectors are marked with shaded regions. The black horizontal dotted lines are the zero demarcation lines.



In Fig. 9d, the regressions show a clear latitudinal shift in the ITCZ due to ENSO. All three datasets exhibit quite similar patterns, with slight variations. There is a positive trend in the American sector, while the trends are negative in the African and Asian sectors. The ERA5 and NCEP exhibit more volatility than the RO, notably in the African and Asian sectors, indicating model dependency in the response to ENSO. Over the Atlantic Ocean, the Ep trends continue to show a negative trend, though less pronounced than the Pacific. The Atlantic's influence on global weather patterns, such as the Atlantic Meridional Overturning Circulation (AMOC) (Mignot and Frankignoul, 2005; Wolter and Timlin, 2011; Menary et al., 2012; Orihuela-Pinto et al., 2022; Liu et al., 2024), can contribute to these trends. Around the Indian Ocean, the Ep trends remain negative but with smaller amplitudes, likely due to monsoon and other regional atmospheric processes. The AMOC is a major cycle that transports warm surface water from the tropics and subtropics in the north to the North Atlantic. The increase in the datasets suggests that this relationship is strong and may be influenced by large-scale ocean-atmosphere interactions such as ENSO. Over the Atlantic Ocean, the coefficients are close to zero, indicating a weaker or more complex relationship between refractivity and Ep. The variation may be due to the role of the Atlantic in climate variability, which can produce heterogeneous climates that are not entirely captured by a single parameter. Measurements in the Indian Ocean show a slight positive trend, albeit with some variability. This suggests a downward correlation between refractivity and Ep, possibly influenced by complex interactions of the region with the monsoonal system and the Indian Ocean Dipole (IOD). The IOD is an 'ENSO'-like phenomenon in the Indian Ocean. ENSO is characterized by anomalous warming (El Niño) or cooling (La Niña) over the equatorial Pacific Ocean, peaking in December (Cherchi and Navarra, 2013; McKenna et al., 2020; Zhang and Han, 2021). Consistent trends in Ep, especially in the America and over the oceans, indicate a significant decrease in atmospheric energy availability, possibly with global climate events such as ENSO, AMOC, and IOD. The relationship between refractivity and Ep varies, with the American part having a stronger relationship and the African and Asian parts having a more complicated relationship.

The MJO (Fig. 9e) has an apparent adverse effect on the ITCZ across the whole longitudes, especially in Africa, where all datasets indicate substantial decreases. In return, the African sector receives the most substantial negative response, where all the datasets agree, indicating the connection between MJO and the ITCZ movement. This result also suggests that the influence of MJO does not only affect equatorial stratospheric GW but also atmospheric refractivity. The MJO is the primary factor influencing the variability in the tropical atmosphere for 30 to 90 days. It interacts frequently with the ocean beneath and impacts several weather and climate systems (Zhang, 2005). The phenomenon involves extensive interconnected patterns of air circulation and convection that move eastward across the Indian and Pacific oceans. MJO is a weather pattern in the lower part of the Earth's atmosphere that features eastward-moving intense thunderstorms and interconnected air movements. Tropospheric convection and wind circulations linked to the MJO influence the propagation of GWs (Li and Lu, 2020) and other atmospheric constituents (Zhang, 2005). A similar result has also been found in Figure 1a of Randel et al. (2021), using the temperature variance to identify the statistical structure of small-scale GWs. Also, large-scale circulations such as the MJO affect the diurnal precipitation cycle, affecting the ITCZ (Kerns and Chen, 2018). The Moss et al. (2016) study presents the first observational evidence that the MJO strongly influences the worldwide variability of stratospheric GWs in the tropics.

The QBO (Fig. 9d) impact, most notably at 50 mbar, shows minor but consistent negative trends across all longitudes. The trends at 30 mbar differ considerably amongst datasets, suggesting a greater sensitivity to the dataset used. Both levels



(30 and 50 mbar) exhibited the most significant influence in Africa, with only minor differences between datasets. The ITCZ
appears less vulnerable to QBO than ENSO and MJO; however, the patterns are still visible. The RO is substantially consistent
with the ERA5 and NCEP, with minor changes. The ERA5 and NCEP data are typically more variable, indicating model or
observational changes. The African sector appears particularly vulnerable to all three oscillations, with the most significant
changes reported in response to ENSO, MJO, and QBO. The American and Asian sectors reacted quite mildly. These findings
emphasize the role of atmospheric oscillations in relocating the ITCZ, with ENSO and MJO having the most significant effect.
The QBO also contributes but with less obvious results. Specific trends are similar across datasets, lending legitimacy to
observed patterns; nonetheless, regional and model-based discrepancies highlight areas for further exploration.

### 3.4 Refractivity and Ep maximum values

To investigate the relationship between refractivity and Ep, we identified the local maxima of zonal refractivity and Ep at
each Gaussian peak, which correspond to the prominent features in the fitted Gaussian profiles, as shown in Fig. 10a,b,c and
b. Fig. 10 illustrates the zonal refractivity and Ep maxima values at these Gaussian peaks for DJF and JJA (a) and for SON
and MAM (b). We note that MAM signifies the transitional months between DJF and JJA, and SON signifies the transitional
months between JJA and DJF. To further capture the linearity between the two parameters (refractivity and Ep), we further
normalized the zonal refractivity and the Ep maxima values at the Gaussian peaks from the result shown in Fig. 10c and d. The
refractivity at different seasons seemed to follow the same trend, especially in the transitional months (Fig. 10). The equatorial
stratospheric GW Ep during DJF and JJA (Fig. 10a) interchange at different locations, while during SON and MAM (Fig. 10b),
the Ep follow the same trend. The results are more apparent for comparison in Fig. 10c and d. Both the refractivity and the Ep
at different seasons follow respective trends. We observed an opposite trend between the refractivity and the Ep (increase in
refractivity and decrease in Ep) during JJA around $60°$W (-60 in Fig. 10c) over tropical South America. The depression in the
GW energy over this region could be due to JJA tropical dynamics (ENSO, for example).
During JJA, refractivity values in the American sector peak for reanalysis (ERA5, NCEP) and RO data. The African sector
experiences fewer refractivity swings than the American sector, with minor seasonal variations. The Asian sector has occasional
peaks, particularly in JJA and MAM, although the shifts are less severe than in the American sector. The most significant
increase in the American sector, notably in RO and NCEP data, occurs during the JJA. DJF showed lower refractivity values
throughout all sectors. MAM and SON depicted minimal refractivity, with MAM slightly increasing in Asia. Ep values in the
American sector drop at around $120°$W and $30°$W, particularly throughout the DJF, SON, and MAM months. The African
sector shows growing Ep values as it advances eastward, with maximums in the DJF and MAM. The Asian area receives
significant increases in Ep, especially between DJF and MAM, indicating higher GW energy. DJF has the highest Ep values
in Asia, showing that GW activity is intense throughout the season. While JJA brings lower Ep values, particularly in the
American sector. SON Ep values were low, with advances in the African sector, but MAM had considerable increases in both
the African and Asian sectors. The American sector has higher refractivity values during JJA, probably due to stronger winds,
which coincide with this region's more significant seasonal differences. During DJF and MAM, the African and Asian sectors
have higher Ep values, possibly due to more significant convective activity in these regions, resulting in higher GW generation.



The differences in refractivity across the RO, ERA5, and NCEP data indicate that these datasets characterize atmospheric parameters differently, especially in areas of high seasonal variability, such as the American sector.

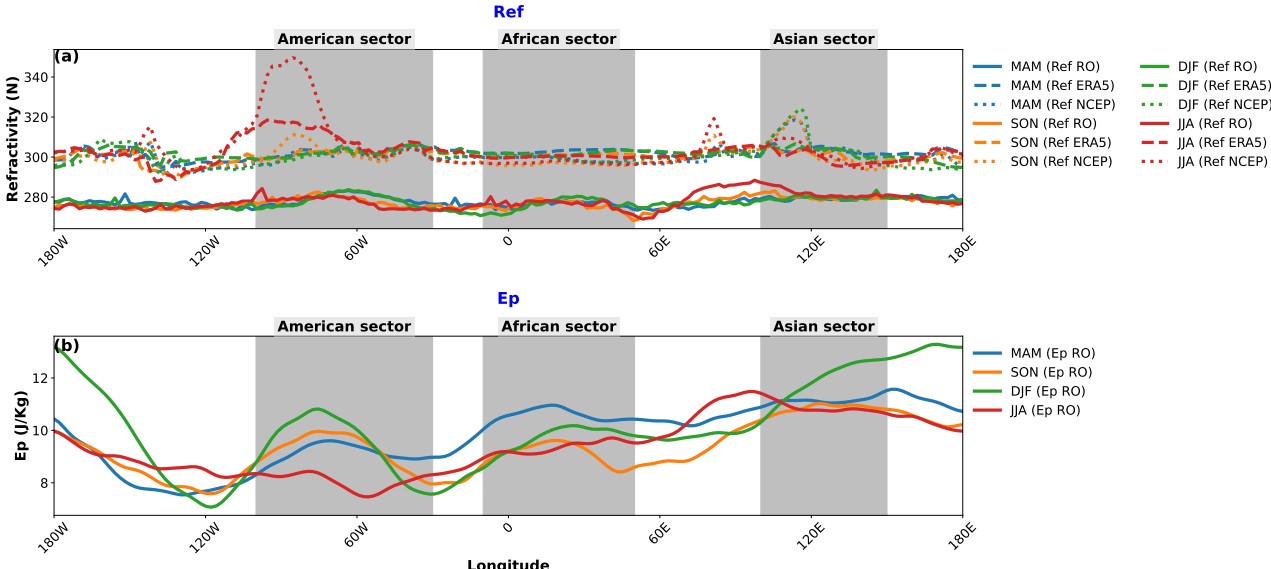

**Figure 10.** The zonal refractivity from RO (solid line), ERA5 (dashed line), NCEP (dotted line), and Ep values at their respective maximum Gaussian. **(a)** The refractivity values at the peak of the Gaussian for DJF (green), MAM (blue), JJA (red), and SON (orange). **(b)** The Ep maxima values are at the peak of the Gaussian for DJF (green), MAM (blue), JJA (red), and SON (orange). The shaded areas mark the American (100°W-30°W), African (10°W-50°E), and Asian sector (100°E-150°E) sectors are marked with shaded regions.

### 3.4.1 *Trends of refractivity and maximum Ep values*

Fig. 11 examines the zonal trends and coefficients of Ep and refractivity values at their Gaussian peak in various longitudinal sectors (America, Africa, Asia) and oceans (Pacific, Atlantic, India) using RO, ERA5, and NCEP data. The trends in refractivity vary spatially but generally fluctuate around zero (Fig. 11a). A slight positive trend is observed for RO and ERA5 in the American sector, with significant variations in NCEP. NCEP shows a slight positive trend, while RO and ERA5 remain near zero over the African sector. Over the ocean, the separation is relatively small, very stable in the Pacific and Atlantic Oceans, and highly variable in the Indian Ocean, which could be influenced by regional factors such as precipitation. The trend of Ep exhibits a consistently negative pattern Fig. 11b. Over the Pacific Ocean, a similar slightly negative trend is observed, associated with broad-scale weather patterns such as ENSO. In Fig. 11c, the American region's relationship between Ep and refractivity values is positive. In contrast, the African and Asian regions show significant changes, making the relationship less clear. Over the Pacific Ocean, the coefficients are generally positive, indicating positive correlations, likely influenced by large-scale ocean-atmosphere interactions such as ENSO. In contrast, the effects over the Atlantic are close to zero, indicating




a weak correlation. At the same time, the Indian Ocean exhibits a positive downward correlation with some variability, which is likely regionally driven in the atmosphere.

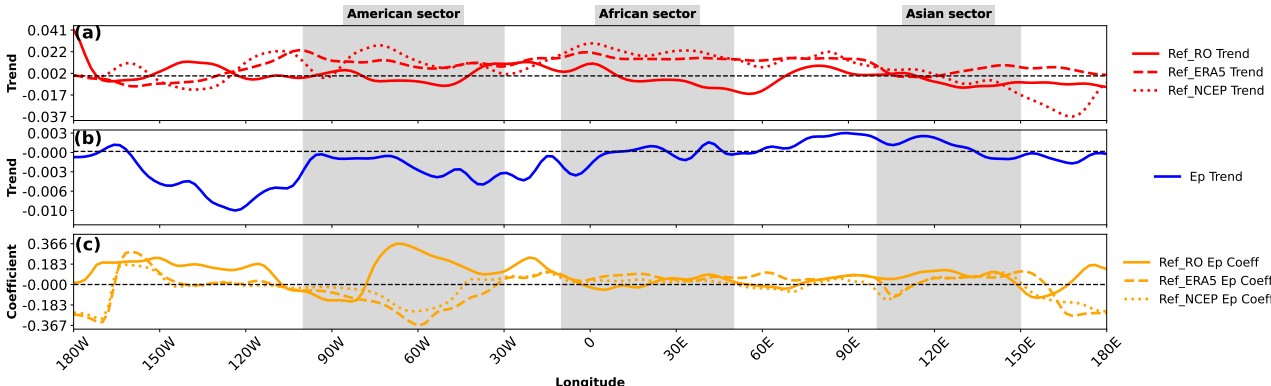

**Figure 11.** The zonal trends and coefficients of the Ep and refractivity from RO, ERA5, and NCEP at their respective Gaussian peak values over 11 years. **(a)** The zonal trends of RO (solid line), ERA5 (dashed line), and NCEP (dotted line) refractivity Gaussian peak values. **(b)** The zonal trends of the Ep Gaussian peak values. **(c)** The zonal coefficients of the Ep and refractivity from RO, ERA5, and NCEP Gaussian peak values. The shaded areas mark the American ($100°$W-$30°$W), African ($10°$W-$50°$E), and Asian ($100°$E-$150°$E) sectors are marked with shaded regions. The black horizontal dotted lines are the zero demarcation lines.

### 3.4.2 *Modes of climate variability and refractivity and Ep maxima values*

The zonal trends of Ep and refractivity from the RO, ERA5, and NCEP Gaussian peak values, each regressed against three significant atmospheric oscillations—ENSO, MJO, and QBO are shown in Fig. 12. The ENSO regression shows positive trends at most longitudes (Fig. 12a), particularly strong over the American sector. The trend increases to the east from $120°$W to $90°$W longitude, indicating that ENSO influences the area, usually the Pacific. In the African and Asian regions, the trend slowly decreases but is still positive, which means the effect of ENSO on GW's intensity is consistent. In Fig. 12b, the MJO

regression showed alternating oscillations from positive to negative. The positive trends are the strongest over the Asian region, specifically for $120°$E to $150°$E. The trend is less pronounced and variable over the American and African sectors, showing variable influences depending on the region and possibly the phase of the MJO. The MJO also has a significant influence, especially in the Indian Ocean and western Pacific region (Asian sector). The QBO effect is present but more subtle and uneven across sectors. These zonal coefficients highlight the different effects of climate variables on GW dynamics at different

latitudes, with ENSO and MJO having the most important effects, especially in the coastal regions where these oscillations are more prominent.

In Fig. 12c, the QBO regressions at 30 mbar (QBO30) and 50 mbar (QBO50) showed smaller trends ($\sim -0.04$ to $-0.05$ J kg$^{-1}$ per month) compared to ENSO and MJO, which show more significant effects of the QBO on GW activity. The QBO30 shows slightly negative trends in the American sector, while the QBO50 has a more oscillatory structure. Over the Asian sector,




especially for 90°E to 120°E longitude degrees, QBO50 exhibits a slightly positive trend, partially indicating an effect in this region. In general, the small magnitudes indicate that although the QBO influences GW strength, the influence is much smaller than that of ENSO and MJO. The most prominent features are related to ENSO, especially in the Pacific region, where the ENSO effect on GW intensity is intense.

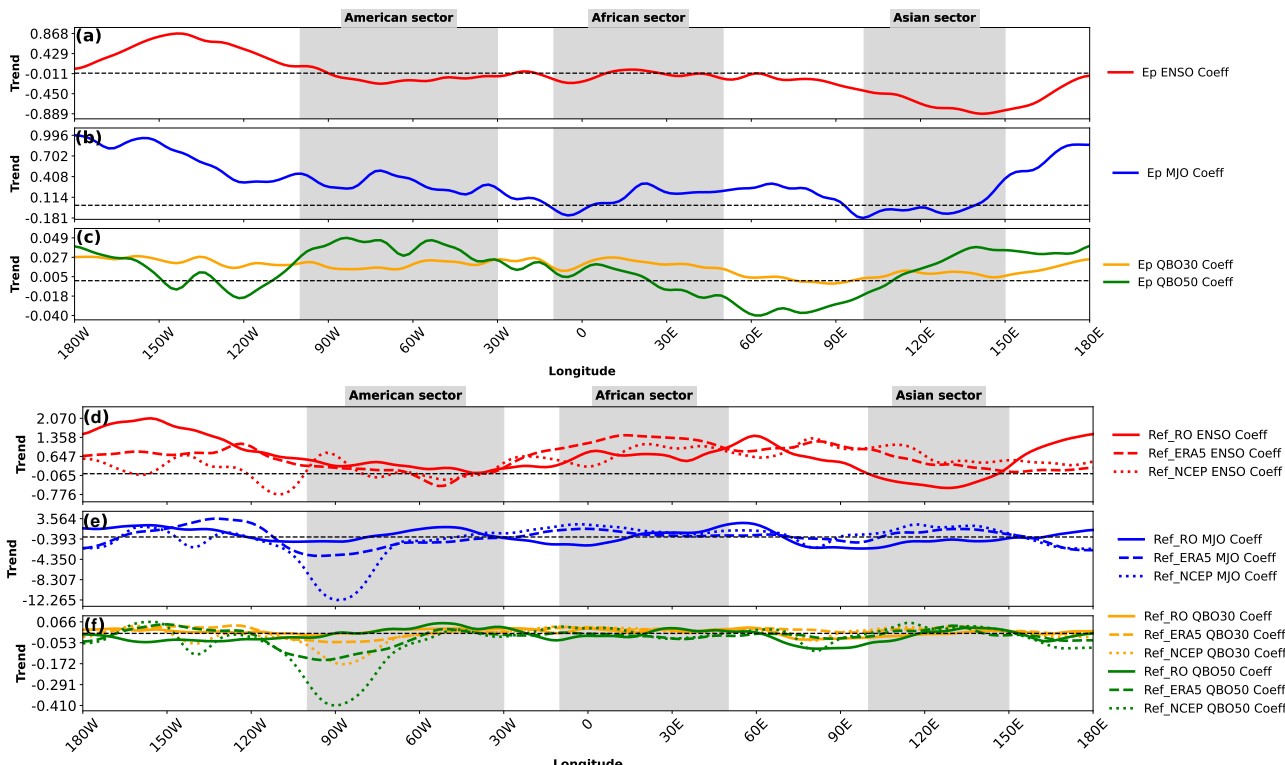

**Figure 12.** The zonal regressions of the atmospheric oscillations with the Ep and refractivity from RO, ERA5, and NCEP at their respective Gaussian peak values over 11 years. **(a-c)** The zonal regressions of the Ep Gaussian peak values with ENSO (**a**, in red line), MJO (**b**, blue line), and QBO at 30 mbar (**c**, orange line) and at 50 mbar (**c**, green lines). **(d-f)** The zonal Refractivity Gaussian peak values by RO (solid lines), ERA5 (dashed lines), and NCEP (dotted lines) regressions with ENSO (**d**, in red lines), MJO (**e**, blue lines), and QBO at 30mbar (**f**, orange line) and 50mbar (**f**, green lines). The shaded areas mark the American (100°W-30°W), African (10°W-50°E), and Asian sector (100°E-150°E) sectors are marked with shaded regions. The black horizontal dotted lines are the zero demarcation lines.

The correlations of refractivity with ENSO showed significant variability in different longitudes (Fig. 12a). The RO data 450  (solid red line) implies high positive trends across the globe; it is seen most markedly over the American and Asian sectors, which means the relationship of ENSO with refractivity is rather direct and firm over these parts of the globe. The ERA5 and NCEP data (Fig. 12d, dashed and dotted red lines) are also observed to follow the same pattern in general but with slightly less pronounced trends than the RO data. The RO data of the Pacific region (150°W–180°W) and the Atlantic region (30°W-60°W) show significant positive trends; the RO data is mainly positive. These observations suggest that the influence of ENSO on these





sea areas might be more significant, which is connected to the immediate impact of the sea-surface temperature anomalies. The MJO-related refractivity trends (Fig. 12e) were found to have a higher variability among different longitudes, in reckon to ENSO, with the RO data having the most significant change. The trends over the American and Asian regions indicate a high deviation between the RO and reanalysis lines, where the RO trends show both positive and negative values. The MJO trends in the Pacific and Atlantic Oceans result in negative values, particularly in the NCEP data (dotted blue lines), meaning that MJO

dampens refractivity in these regions. The QBO-motivated trends, represented in Fig. 12f as $30\,\mathrm{mbar}$ and $50\,\mathrm{mbar}$, point out that QBO has the lowest trend magnitudes over the longitudes ($80°\mathrm{W}$–$100°\mathrm{W}$), whereas ENSO and MJO show higher trends. The only consistent trends for the RO, ERA5, and NCEP datasets are their slight positive and negative variations around the globe. Over the oceans, particularly in the Pacific and Atlantic, most of the trends remain neutral or slightly negative, which means the effect of QBO on the refraction in these areas is almost zero.

The American, African, and Asian regions show distinct sensitivity to MJO and QBO, and the Americas and Asian regions show more pronounced responses to ENSO, especially in RO data. ENSO effects are most pronounced in oceanic regions, especially in the Pacific Ocean, indicating a strong correlation between sea surface temperature and atmospheric divergence. This finding is consistent with the well-known effects of ENSO on the ocean and atmospheric interactions. The MJO shows distinct effects over the ocean, which can produce cold water in some areas, especially the Pacific. The variability could be

due to the transient diffusive nature of the MJO phases interacting with oceans (Klotzbach et al., 2019). On the other hand, QBO exhibits a more negligible effect in oceanic regions, suggesting that its effect on refractivity is more pronounced in the troposphere rather than the stratosphere where ocean-atmosphere interactions are stronger. RO data often show strong trends, possibly due to the higher resolution and sensitivity of GNSS RO measurements compared to reanalysis datasets ERA5. NCEP data typically show weaker trends, possibly due to assimilation of observational data and model constraints.

## 4   Discussion

This study updated the use of RO refractivity data to find the ITCZ and used COSMIC-2 measurements to show the relationship between the ITCZ and the equatorial stratospheric GW. Refractivity represents the shape of the ITCZ distribution across latitudes and provides a more accurate identification of the ITCZ than other variables, such as precipitation and vertical wind. Therefore, this technique is considered appropriate for the simulated geographic distribution of ITCZs in equatorial

stratospheric GW generation. Based on our findings, the equatorial stratospheric GW closely follows the path of the ITCZ. Few studies (e.g., Straub and Kiladis (2002), Ratnam et al. (2004), Dias and Pauluis (2009)) have found a good relationship between the features often used as proxies to identify the ITCZ and the equatorial stratospheric GW. For example, Ratnam et al. (2004) found a good correlation between the OLR and equatorial GWs in the lower stratosphere. Dias and Pauluis (2009) studied the dynamics of convectively connected GWs traversing the precipitation region using an ideal model for large-scale

atmospheric rotation and found that Kelvin waves propagate along the width of the ITCZ in the Rossby radius. Kelvin waves propagate at the speed of the GW ($\sim 15\,\mathrm{m\,s^{-1}}$), while the propagation speed of a narrow ITCZ is comparable to the speed of



a dry GW ($\sim$50 m s$^{-1}$). Straub and Kiladis (2002) confirmed that the propagation characteristics of equatorial trapped Kelvin waves have revealed evidence of convective activity propagating through the central axis of the East Pacific ITCZ.

The results show that the global ITCZ exhibits a significant and abrupt seasonal movement from DJF to JJA. A thorough
analysis reveals that the equatorial jumps of the regional ITCZ over oceanic regions like the Indian Ocean and the western Pacific are the primary cause of the abrupt migration in JJA. In contrast, contributions from South America and the western and central Pacific are the primary cause of the migration in DJF. In contrast to the Basha et al. (2015) results show that there is no dramatic movement of the regional ITCZ over Africa, the eastern Pacific, or the Atlantic regions, our results showed a northward shift during JJA/SON and an equatorward shift during DJF/MAM over Africa and the Indian Ocean. This result was
also evident in the E$_p$ maxima, which showed the most significant seasonal latitudinal difference. As mentioned earlier, the ITCZ follows, and the trade winds converge further in the SH, resulting in clouds with deep convective systems that force the ITCZ to move southward. In JJA, the ITCZ returns, and the trade winds concentrate in the NH. Our findings on the location of the ITCZ are consistent with those of previous studies by Gu and Zhang (2002); Hu et al. (2007); Läderach and Raible (2013); Byrne et al. (2018).

The results in Fig. 8d show a high global variability near the ITCZ and that GW E$_p$ shows regional differences in climate dynamics. Collecting data to reveal the complexities of these approaches entirely is essential, as evidenced by the different trends and ratios between the datasets. The link between ITCZ activity and GW changes in Africa and Asia could result from significant regional phenomena like deep convective and monsoon activities, as indicated by the significant coefficients in these sectors. Understanding these relationships is essential to our ability to do more in the ITCZ and improve climate conditions.
MJO, ENSO, and QBO directly affect the position and strength of the ITCZ (Kerns and Chen, 2018; Reed et al., 2019), and the effects on wind and temperature. The ITCZ shifts southward during El Niño events and northward during La Niña (Zhang and Han, 2021). These changes in the location of the ITCZ affect the formation and propagation of GWs, which are often initiated by the ITCZ and convective activity. Moreover, the vertical radiation associated with these oscillations may affect the growth and propagation direction of the GW (Nath et al., 2015; Kang et al., 2020; Randel et al., 2021; Essien et al., 2022; Holt et al.,
2022). Understanding these deep interactions is critical for a better understanding of tropospheric and stratospheric two-way interactions.

The ENSO can influence the relationship between the ITCZ and GW through various mechanisms (e.g., Atmospheric Circulation Changes, Ocean-Atmosphere Interaction, Changes in Convection, Wave-Mean Flow Interaction, Stratosphere-Troposphere Coupling). At the time of El Niño occurrences, the Walker circulation—an east-west atmospheric circulation along the equatorial Pacific, characterized by low-level trade winds converging towards the western Pacific and upper-level re-
turn flow—weakens. This weakening causes shifts in the position and strength of the ITCZ, leading to changes in precipitation conditions and wind patterns as GW changes (Xie et al., 2018). El Niño events also increase activity, intensifying cloud formation (Zhou et al., 2024). Increased convective activity, specifically the rising motion of warm air masses, contributes to the structure and strength of GWs that affect the ITCZ. According to Xie et al. (2018), ENSO-related variations in deep convection
are mostly confined north of the equator, a meridional pattern not effectively captured by averages in the equatorial-centered ENSO region. This imbalance in ENSO atmospheric anomalies is caused by the meridional asymmetry of the eastern Pacific





climate, in which the south of the equator is typically below the convection threshold, and the ITCZ is shifted north of the equator (Xie et al., 2018). According to Moss et al. (2016), the MJO might influence stratospheric GWs in two main ways. First, the MJO's convection modulation may affect GW excitation. Second, MJO tropospheric wind anomalies could influence wave propagation through critical-level filtering. Thus, these mechanisms may affect stratospheric GWs and atmospheric refractivity. Few studies have examined how the MJO affects stratospheric GWs. Also, Horinouchi (2008) predicted that the lower convection during the MJO's inactive phase would yield higher stratospheric GW energy and momentum fluxes than the active phase because of better vertical propagation of the longer vertical wavelengths.

## 5 Conclusions

In this study, we determined the global ITCZ from RO refractivity (COSMIC and METOP satellites) and ERA5 and NCEP reanalysis data, as well as Ep maxima in the equatorial region using the Gaussian fitting method from 2011 to 2021. Using these estimates, we investigated the trends and relationships between the ITCZ and GW Ep maxima locations for variations across different longitudinal sectors and the effects of modes of climate variability such as ENSO, MJO, and QBO. The ITCZ position shifts by about $10°$ in DJF and JJA and by about $5°$ in SON and MAM over the ocean regions globally. Generally, the ITCZ derived from RO and reanalysis data shows a similar pattern. The position and strength of the ITCZ vary from year to year. The interannual variability has been about $\sim 5°$ to the north and south. The global equatorial stratospheric GW Ep maxima showed seasonal changes similar to the ITCZ. The global equatorial stratospheric GW Ep maxima were more consistent in the NH and SH and had a much smaller gap than the ITCZ. The Ep position changes by $5°$ during DJF, JJA, and between SON and MAM. The Ep maxima in DJF and JJA suggest two sites of convergence over the South American Amazon and the equatorial Pacific. Our results show that the ITCZ positions are consistent with other features such as specific humidity, OLR, and vertical wind data. Our results showed good agreement between OLR and peak-specific humidity, especially at DJF. The refractivity and Ep maxima exhibited opposite trends in certain regions, notably during JJA over tropical South America, where increased refractivity coincided with decreased Ep. The ITCZ derived from RO and reanalysis data shows different patterns in trends at different longitudinal geolocations and the refractivity values at the peak of the Gaussian. The ITCZ showed positive trends in its latitudinal position, with variability across different longitudinal sectors. The correlation between ITCZ and Ep positions was higher in the African and Asian sectors, indicating a solid link between these parameters. The zonal correlation between ITCZ and GW energy was positive overall, especially in the Asian monsoon regions, implying that convective systems on a mesoscale level are significant sources of GWs in those regions. The ENSO affected Ep and ITCZ locations differently in different subregions. Positive ENSO phases brought about a north shift of Ep in the Asian sectors and a southward shift of these locations in the American and African sectors. The ITCZ also showed a positive trend in the American sector, while negative trends were revealed in the African and Asian sectors. The MJO consistently negatively influenced the Ep maxima locations across all sectors, with a southward shift being more pronounced in the African and Asian sectors. The MJO also strongly impacted the ITCZ, especially in the African sector, indicating a substantial negative influence on atmospheric refractivity and GW activity. The QBO's influence was less relevant than that of the ENSO and MJO, with the QBO showing



a negative effect that was consistent across all longitudes, albeit more mild at $50$ mbar. The region of Africa showed the most significant sensitivity to QBO, and the ITCZ showed some sensitivity to this oscillation. The variations showed that the refractivity peaked positively in the American sector during JJA. In contrast, Ep values were highest in the African and Asian sectors during DJF and MAM, likely during convective activities that increased. The study reveals the different ENSO, MJO, and QBO modulations on atmospheric refractivity over different longitude sectors and distinct continents. ENSO showed a

more significant contribution to ENSO, especially over oceans.

*Data availability.* CDAAC, ECMWF, and NOAA exclusively provide the data used in this study, and they were obtained from http://cdaac-www.cosmic.ucar.edu/cdaac, https://www.ecmwf.int/en/forecasts/dataset/ecmwf-reanalysis-v5, https://psl.noaa.gov/data/gridded/data.ncep.reanalysis.html, respectively.

*Author contributions.* Conceptualisation, Ayorinde, T. T., Wrasse, C. M., and Takahashi H.; methodology, Ayorinde, T. T.; software, Ay-
orinde, T. T.; validation, Wrasse, C. M., and Takahashi H., Luis Fernando Sapucci; formal analysis, Ayorinde, T. T.; investigation, Ayorinde, T. T.; resources, CDAAC; data curation, Ayorinde, T. T.; writing—original draft preparation, Ayorinde, T. T.; writing—review and editing, Wrasse, C. M., Takahashi H., Luis Fernando Sapucci, Figueiredo, C. A. O. B.; visualisation, Barros D. and, Essien P.; supervision, Wrasse, C. M., and Takahashi H.; project administration, Ayorinde, T. T.; funding acquisition, Wrasse, C. M.

*Competing interests.* The authors declare that there are no competing interest.

*Acknowledgements.* The authors acknowledged CDAAC, ECMWF, and NOAA for providing the data. We also acknowledge the financial support provided by the Brazilian Ministry of Science, Technology, and Innovations (MCTI) and the Brazilian Space Agency (AEB) underf the grant number 20VB.0009 and the Conselho Nacional de Desenvolvimento Científico e Tecnológico (CNPQ) under the process number $141373/2019 - 9$, $303871/2023 - 7$. Thanks to Fundação de Apoio a pesquisa do estado da Paraíba (FAPESQ) under the process number $2021/04696 - 6$. Thanks to Fundação de Apoio a pesquisa do estado da Sao Paulo (FAPESP)





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
