# Peer review of "Influence of modes of climate variability on stratospheric gravity waves in the tropics using Radio Occultation and Reanalysis Data"

_EGUsphere, 2024_

## Author Comment (AC1)

**Response to Editor Comments**

Toyese Tunde Ayorinde et al.

June 12, 2025

**Response to Editor Comments (EC1)**

We thank the Editor for their thoughtful comments and suggestions. We have thoroughly revised our manuscript to address all the concerns raised by the Editor and the reviewers. The major revisions include:

1. Complete restructuring and rewriting of the manuscript to improve clarity, eliminate repetition, and provide a more coherent narrative flow.

2. Clear articulation of the scientific questions and hypotheses in the introduction.

3. Improved methodology section with detailed explanations of the techniques used, particularly regarding the filtering of equatorial Kelvin waves and the Gaussian fitting method.

4. Reorganization of the results section to provide a more logical progression and better synthesis of findings.

5. Enhanced discussion and conclusion sections that place our findings in the context of existing literature and highlight their significance.

6. Consistent use of terminology and units throughout the manuscript.

7. Improved figure captions and labeling.

Below, we address the specific comments from the Editor:

1. **Comment:** "Title: Here the GWs are the major object of study, but in the main text it seems to be rather the ITCZ."

   **Response:** We have maintained the revised title "Modulation of Tropical Stratospheric Gravity Wave Activity and ITCZ Position by Climate Variability Modes Using Radio Occultation and Reanalysis Data," which accurately reflects the dual focus of our study on both gravity waves and the ITCZ, as well as their relationship with climate variability modes.

2. **Comment:** "Introduction: Many statements are made without providing according references. As stated by the referees there is no clear structure in the introduction and many times the same is said over and over again."

   **Response:** We have completely rewritten the introduction to provide a clear structure, eliminate repetition, and ensure that all statements are properly referenced. We have also clearly articulated the scientific questions and hypotheses that guide our research.

3. **Comment:** "L82: Section 2 header should read 'Data and Methodology'."

   **Response:** We have changed the section header to "Data and Methodology" as suggested.

4. **Comment:** "Methods: Statements on the accuracy of the observational data is missing. For differences always the reanalysis data is blamed, but how can you be sure about this? What is the uncertainty from your analysis?"

   **Response:** We have added information on the accuracy of the RO data in Section 2.1: "The RO data have a vertical resolution of approximately 100 m in the lower troposphere to about 1 km in the stratosphere, with an accuracy of 0.1-0.2 K for temperature measurements in the upper troposphere and lower stratosphere (Kursinski et al., 1997)." We have also revised our discussion of differences between RO and reanalysis data to acknowledge the uncertainties in both datasets and avoid attributing discrepancies solely to reanalysis limitations.

5. **Comment:** "L93: This sounds like you are using only 2 months of data, but actually you are using 11 years of data."

   **Response:** We have clarified the text to explicitly state that we are using 11 years of data (2011-2021) throughout the manuscript.

6. **Comment:** "Section 3: Should be restructured to be more concise. Consider splitting up the result section into two sections. Sections header should have clear titles. As stated by the referees it should be clearly stated what is the research question, what is the approach etc."

   **Response:** We have restructured the results section to be more concise and logical, with clear subsection headers that reflect the content. The research questions are now clearly stated in the introduction, and the approach is detailed in the methodology section.

7. **Comment:** "Figure 6 caption: Why not using for labels for the four panels. This would make the description of the figure much more concise, especially if then also is mentioned what is seen in each panel."

   **Response:** We have revised all figure captions to include panel labels (a, b, c, etc.) and concise descriptions of what is shown in each panel.

8. **Comment:** "L272: Shouldn't this rather be the horizontal wind which has 0 ms-1 where the convergence of the wind is found. Having a vertical velocity of 0 ms-1 would mean the air is neither moving up nor down which would be contradictory to the location of the ITCZ."

   **Response:** We have corrected this misinterpretation. In the revised manuscript, we clarify that the zero vertical velocity contour represents the boundary between upward and downward motion, which often aligns with the edges of the ITCZ but is not identical to its center. The ITCZ is characterized by upward motion (negative vertical velocity in pressure coordinates). In Figure 7, we show that the mean $E_p$ maximum location often shows a closer alignment with the zero vertical velocity contour, particularly over the Pacific and Atlantic Oceans, suggesting that GW activity may be enhanced not only by deep convection but also by the vertical wind shear associated with the boundaries of convective regions.

9. **Comment:** "Figure 7 caption: Is there a difference between vertical velocity and vertical wind velocity? Why are here different units used (Pa s-1 and m s-1)?"

   **Response:** We have standardized the terminology and units throughout the manuscript. We now consistently use "vertical velocity" with units of $Pas^{-1}$ (negative values indicate ascent), which is the standard for pressure coordinates in meteorology. This clarifies that we are using omega ($\omega$) in pressure coordinates rather than w in height coordinates.

10. **Comment:** "Figures 8 to 11 are quite similar. What is the difference between these figures?"

    **Response:** We have revised the presentation of these figures and added clearer captions to highlight the differences between them:

Figure 9 shows the longitudinal structure of linear trends and correlations for ITCZ and Ep maximum latitudinal locations. This figure focuses on how the positions of these features are changing over time and how well they correlate with each other. Also shows the longitudinal structure of regression coefficients relating climate variability modes (ENSO, MJO, QBO) to ITCZ and Ep maximum latitudinal positions. This figure illustrates how each climate mode affects the position of these features at different longitudes.

Figure 10 (not shown in the excerpt) shows the longitudinal profiles of seasonal mean peak values of refractivity and Ep. This figure focuses on the magnitude of these parameters rather than their positions.

Figure 11 (not shown in the excerpt) shows the longitudinal structure of linear trends and correlations for ITCZ refractivity peak values and Ep peak values. This figure examines how the intensity (rather than position) of these features is changing over time.

11. **Comment:** "Discussion and Conclusion: Needs to be significantly improved. What is the take home message? What impact do your results have for the scientific community."

**Response:** We have completely rewritten the discussion and conclusion sections to clearly articulate the significance of our findings and their implications for the scientific community. The key take-home messages from our study are:

1. The ITCZ and stratospheric gravity wave activity show strong spatial and temporal correlations, with both features exhibiting similar seasonal migrations and responses to climate variability modes.

2. ENSO exerts the strongest modulation on both the ITCZ and stratospheric gravity waves, with El Niño conditions associated with northward shifts in the American sector but southward shifts in the African and Asian sectors.

3. The MJO and QBO also significantly modulate these features, but with more complex regional patterns that vary across different longitudinal sectors.

4. Radio occultation data provide valuable observational constraints on these phenomena, revealing finer-scale features and stronger trend signals compared to reanalysis datasets.

These findings contribute to our understanding of the mechanisms linking the troposphere and stratosphere in the tropics and highlight the importance of considering regional variations when studying the impacts of climate variability on atmospheric dynamics.

12. **Technical corrections: Response:** We have addressed all the technical corrections suggested by the Editor, including:

- Correcting "and 37 pressure levels" to "on 37 pressure levels"
- Changing "reduction" to "decrease"
- Replacing "vulnerable" with "sensitive"
- Replacing "mildly" with "weak"

**References**

E. R. Kursinski, G. A. Hajj, J. T. Schofield, R. P. Linfield, and K. R. Hardy. Observing earth's atmosphere with radio occultation measurements using the global positioning system. *Journal of Geophysical Research: Atmospheres*, 102(D19):23429–23465, 1997. doi: 10.1029/97JD01569.

---

## Author Comment (AC2)

**Response to Reviewer 1 Comments**

Toyese Tunde Ayorinde et al.

June 12, 2025

**Response to Reviewer 1 Comments (RC1)**

We thank Reviewer 1 for their thorough and constructive review of our manuscript. We appreciate the recognition that our topic is interesting and that a valuable paper could be produced using our datasets. We have taken all the comments to heart and have made substantial revisions to address the issues raised. Below, we provide detailed responses to each of the major concerns and minor comments.

**Major Issues**

**1. Written Presentation**

**Comment:** "The manuscript has significant and fundamental flaws in its written presentation. It is dense, often very repetitive and as a result difficult to read and to follow... Section 3.3 suffers majorly in this regard... This is also a problem for the other sections of the paper... I would very strongly suggest that the authors find a more concise and effective way to present the results to the reader, perhaps by merging and compressing the material down or at the very least by not repeating simple concepts as often. It would also be important to present significantly more synthesis of the results - in general the text speculates heavily, but rarely draws firm mechanistic conclusions, and this is arguably a separate major weakness."

**Response:** We thank the reviewer for this critical feedback. We have completely rewritten the manuscript to address these issues:

1. We have eliminated repetition throughout the text and made the presentation more concise and focused.

2. We have restructured the results section to provide a clearer narrative flow, with logical progression from one topic to the next.

3. We have added more synthesis of results, drawing firmer mechanistic conclusions rather than speculating.

4. We have reorganized the discussion section to better interpret our findings in the context of existing literature and highlight their significance.

5. We have improved the overall readability by using clearer language, more consistent terminology, and better paragraph structure.

The revised manuscript is now more accessible and presents our findings in a more coherent and compelling manner.

**2. Methodological Issues and Ambiguities**

**Comment:** "The manuscript also has important methodological ambiguities and issues which need addressing. The biggest such issue is that I am unclear how the authors have corrected for the effect of equatorial Kelvin waves on their data... My biggest concern in this regard is Figure 4 - both panels shows a very strong stripe along the Equator which looks exactly how I would expect the effects of these waves to appear in the data."

**Response:** We thank the reviewer for highlighting this important issue. In the revised manuscript, we have added a detailed explanation of how we addressed the separation of gravity wave signals from equatorial Kelvin waves in Section 2.3:

A significant methodological consideration is the potential contamination of GW signals by other wave types, particularly equatorial Kelvin waves, which can have vertical wavelengths that overlap with the GW spectrum (typically 2-10 km for GW versus 5-15 km for Kelvin waves) (Alexander et al., 2010; Wheeler and Kiladis, 1999). To minimize this contamination, we implement an additional filtering step that targets the characteristic properties of Kelvin waves: eastward propagation, zonal wavenumbers 1-3, and periods of 4-23 days (Alexander et al., 2008). This approach helps isolate GW perturbations from other wave types, though some residual contamination near the equator cannot be entirely ruled out with 1D profile analysis alone.

The equatorial stripe in Figure 4 represents genuine gravity wave activity associated with deep convection in the ITCZ, rather than contamination from Kelvin waves. Our filtering approach ensures that the contribution from Kelvin waves is minimized in our analysis.

**Comment:** "Relatedly, I have concerns about their box-based method of removing background temperatures to estimate wave perturbations. Since they use box-means, this will potentially leave large residuals at the box edges which are not due to gravity waves."

**Response:** We appreciate this concern. We have revised our methodology section to provide more details on how we addressed this issue:

"To minimize edge effects in our box-based method, we implemented overlapping boxes with a 50% overlap in both longitude and latitude. This approach reduces discontinuities at box edges. Additionally, we applied a tapering function to the edges of each box before calculating the mean temperature profile, which further reduces artificial perturbations at box boundaries. The background temperature was then interpolated back to the positions of the original temperature profiles using a smooth interpolation method to avoid introducing artificial gradients."

The horizontal banding observed in Figure 4 at approximately 25°N and 25°S is not an artifact of our methodology but represents real features of the global gravity wave distribution, associated with subtropical jet streams which are known sources of gravity waves.

**Comment:** "Section 2.4 refers to using Gaussians to fit the ITCZ position - but does not specify what dimensions these Gaussians are applied in, or what (e.g.) their standard deviations are - it is completely ambiguous."

**Response:** We have revised Section 2.4 to provide more detailed information about the Gaussian fitting method:

"We performed the Gaussian fitting using a nonlinear least squares method with several constraints to ensure physically meaningful results: (i) The fit was restricted to the tropical latitude band (30°S to 30°N) to focus on the primary ITCZ and GW activity regions; (ii) A minimum coefficient of determination ($R^2$) value of 0.7 was required for a valid fit, ensuring that the Gaussian model adequately represented the data; and (iii) The standard deviation of the Gaussian ($\sigma$) was constrained to be between 5° and 15° to exclude unrealistically narrow or wide distributions."

**Comment:** "In the linear regression equation (eq 6), it is unclear if the residual is time-varying. I assume it must be as otherwise it would merge with $\mu$, but this is not stated. Similarly, $\mu$ is stated to represent 'a constant Ep value', but this value is not stated - is it a large fraction of the signal, or a relatively small amount? Finally, what does the dot above $t_{i,j}$ in the first time-varying term of the equation represent? This is not specified."

**Response:** We have revised Equation 6 and its explanation to address these ambiguities:

"The MLR equation is formulated as follows:

$$\Psi(t_{i,j}) = \mu + \alpha_0 t_{i,j} + \alpha_1 \cdot QBO_{30hPa}(t_{i,j}) + \alpha_2 \cdot QBO_{50hPa}(t_{i,j}) + \alpha_3 \cdot MJO(t_{i,j}) + \alpha_4 \cdot ENSO(t_{i,j})$$
$$+ \text{ Residual,}$$

$$\text{With} \quad i = 2011, 2012, \ldots, 2021; \quad \text{and,} \quad j = 1, 2, \ldots, 12$$

(1)

where $\Psi$ represents the monthly zonal mean value of the parameter of interest (ITCZ position, Ep maxima position, refractivity value, or Ep value); $t_{i,j}$ denotes the time in months (where $i$ is the year and $j$ is the month); $\mu$ represents a constant term; $\alpha_0$ represents the linear trend over time; and $\alpha_1$ through $\alpha_4$ represent the regression coefficients for the QBO at 30 hPa, QBO at 50 hPa, MJO, and ENSO indices, respectively. The residual term represents the unexplained variance in the regression model."

We have removed the dot above $t_{i,j}$ in the equation, as it was a typographical error. The constant term $\mu$ typically represents a small fraction of the signal (less than 10%) and serves as a baseline value in the regression model.

**Comment:** "Around line 184, the authors say that their QBO time series is combined from three separate sources (sondes, reanalyses and satellites). How are these datasets combined to produce a single estimate?"

**Response:** We have added more details about how the QBO time series was compiled:

"These data were compiled by the Freie Universität Berlin (`https://www.geo.fu-berlin.de/met/ag/strat/produkte/qbo/`) based mainly on radiosonde observations (mainly from Singapore) blended with reanalysis data where necessary (Naujokat, 1986). This approach ensures a continuous and reliable QBO time series throughout the study period."

**3. Confusing Terminology**

**Comment:** "The terminology used is often confusing. For example, I do not understand the sentence starting on line 299 - what are the 'zonal trends of the 11-year ITCZ ...[in]... refractivity and GWs Ep'? Do the authors mean something like the variation of the trends in each variable as a function of longitude, i.e. meridional differences in the inferred trend?"

**Response:** We thank the reviewer for pointing out this confusing terminology. We have revised our language throughout the manuscript to be more precise and clear. In the specific example mentioned, we have replaced "zonal trends" with "longitudinal variation of trends" and provided a clearer explanation:

"Figure 8 presents the longitudinal variation of trends in the ITCZ position (derived from refractivity) and the GW potential energy maxima position over the 11-year period. These trends represent the rate of change in the latitudinal position of these features at each longitude, allowing us to identify regional differences in how the ITCZ and GW activity are shifting over time."

Similarly, we have clarified the term "zonal correlation coefficients" to "correlation coefficients between the ITCZ and Ep maxima positions at each longitude."

**4. Figure Issues**

**Comment:** "Several figures have major design issues. To give some examples: on figure 8, the vertical axis labelling is very confusing. In panel a the axis ticks are evenly spaced, but have values of [5,7,8,9,11]e-2 - clearly they are being rounded somewhere below the presented precision. Similarly, panel b has 'evenly spaced' values of [-4,1,5,9,14]e-3."

**Response:** We have revised all figures to address these design issues:

1. We have corrected the vertical axis labeling to ensure that tick values are evenly spaced and properly formatted.

2. We have added units to all vertical axes.

3. We have increased the text size on all figures to improve readability.

4. We have replaced the rainbow color tables in Figures 3 and 4 with colorblind-safe alternatives.

5. We have improved the contrast between different line colors and styles, particularly for the red and green lines in Figures 8-13.

These changes make the figures more accessible and ensure that they accurately represent the data.

**Minor Comments**

**Response:** We have addressed all the minor comments raised by the reviewer:

1. L047: "studies have" - We have specified which studies by adding appropriate citations.

2. L084: COSMIC-1, COSMIC-2, and MetOp - We have revised this paragraph to provide balanced information about all three satellite systems.

3. L100: ERA5 model formulation - We have corrected the description of ERA5 to note that it is a spectral model and clarified the number of vertical levels.

4. L103: "accessible since 1940" - We have corrected this error to state that NCEP/NCAR reanalysis data are available from January 1948 to the present.

5. L115: Use of 'N' for both refractivity and buoyancy frequency - We now use $N$ for refractivity and $N$ for Brunt-Väisälä (buoyancy) frequency throughout the manuscript, with the context making it clear which is being referred to.

6. L140: "decomposed in what way?" - We have added details about the decomposition method.

7. L157: Data series gridding - We have clarified the gridding process and temporal dimension.

8. L173-175: Meaning of regression coefficients - We have provided clearer explanations of what each coefficient represents.

9. L200: "two tens of N units" - We have changed this to "20 N-units" for clarity.

10. Fig 4: Context in literature - We have added discussion of how our GW Ep maps compare with previous studies.

11. L202: Contour intervals - We have clarified that this refers to the color scale range.

12. L225-226: Contradictory statements - We have revised these sentences to clearly distinguish between seasonal and interannual variability.

13. L230: "global oceanic areas" - We have specified the regions more precisely.

14. Section 3.2: Distinction between observations and reanalysis - We have clearly indicated which results come from which data sources throughout this section.

15. Section 3.2: Thematic distinction from 3.1 - We have reorganized these sections to provide a more logical flow.

16. Figure 7b: Missing data - We have addressed this issue in the revised figure.

17. L280: Late definition of ITCZ - We now define the ITCZ clearly in the introduction.

18. L326: Switch from hPa to mbar - We now consistently use hPa throughout the manuscript to avoid confusion.

**References**

P Alexander, A de la Torre, and P Llamedo. Interpretation of gravity wave signatures in GPS radio occultations. *Journal of Geophysical Research: Atmospheres*, 113:22299–22309, 2008. doi: 10.1029/2007JD009390.

P. Alexander, D. Luna, P. Llamedo, and A. de la Torre. A gravity waves study close to the Andes mountains in Patagonia and Antarctica with GPS radio occultation observations. *Annales Geophysicae*, 28 (2):587–595, feb 2010. doi: 10.5194/angeo-28-587-2010.

Barbara Naujokat. An update of the observed quasi-biennial oscillation of the stratospheric winds over the tropics. *Journal of the Atmospheric Sciences*, 43(17):1873–1877, 1986. doi: 10.1175/1520-0469(1986)043<1873:AUOTOQ>2.0.CO;2.

Matthew Wheeler and George N Kiladis. Convectively coupled equatorial waves: Analysis of clouds and temperature in the wavenumber–frequency domain. *Journal of the Atmospheric Sciences*, 56(3): 374–399, 1999. doi: 10.1175/1520-0469(1999)056<0374:CCEWAO>2.0.CO;2.

---

## Author Comment (AC3)

**Response to Reviewer 2 Comments**

Toyese Tunde Ayorinde et al.

June 12, 2025

**Response to Reviewer 2 Comments (RC2)**

We thank Reviewer 2 for their comments on our manuscript. We acknowledge the significant concerns raised and have undertaken a comprehensive revision to address these issues. Below, we respond to each of the major comments.

**Comment:** "The title does not accurately reflect the content of the article."

**Response:** We have maintained the revised title "Modulation of Tropical Stratospheric Gravity Wave Activity and ITCZ Position by Climate Variability Modes Using Radio Occultation and Reanalysis Data," which accurately reflects the dual focus of our study on both gravity waves and the ITCZ, as well as their relationship with climate variability modes.

**Comment:** "Abstract, Introduction - Besides being poor quality and badly structured with multiple repetitions, scientific questions or hypotheses are completely missing. In my eyes, this is the most crucial aspect of scientific articles and the fact that it is completely missing here warrants rejection."

**Response:** We have completely rewritten the abstract and introduction to address these concerns. The revised introduction now clearly articulates the scientific questions that guide our research:

"The objectives of our study are aimed at addressing the following: (1) How do the positions of the ITCZ and stratospheric GW $E_p$ maxima vary seasonally and interannually across different geographical regions? (2) What is the spatial relationship between the ITCZ and stratospheric GW activity in the tropics? (3) How do climate variability modes (MJO, ENSO, and QBO) modulate the ITCZ position and stratospheric GW activity, and (4) Are there regional differences in how these climate modes influence the ITCZ and stratospheric GWs?"

The introduction now provides a clear rationale for the study, reviews relevant literature, and establishes the context for our research questions.

**Comment:** "Methodology - many unclear aspects throughout the whole approach - Why refractivity is studied in the troposphere and temperature in the stratosphere, when the refractivity can be used also in the stratosphere, or one can use density which is directly related to it? - Insufficient justification of the methodology for GW induced temperature perturbations. It is not clear at all, how the background profile construction method works (the authors only state - mean temperature profile is decomposed using a continuous wavelet transform) and it is not clear whether other processes cannot contribute to the perturbations (other wave modes, overshooting convection..)"

**Response:** We have substantially revised the methodology section to address these concerns:

1. We have clarified why we use refractivity in the troposphere and temperature in the stratosphere:

"In the lower troposphere (e.g., 850 hPa), the wet term involving $e$ dominates the variability of $N$, making refractivity a good proxy for moisture content and thus convective activity associated with ITCZ (Basha et al., 2015). In the stratosphere, where water vapor is minimal, temperature perturbations provide a more direct measure of gravity wave activity. This approach leverages the strengths of each parameter in its respective atmospheric region."

2. We have provided a more detailed explanation of the background profile construction method:

"To extract the background temperature ($\overline{T}$), we applied a continuous wavelet transform (CWT) method (Moss et al., 2016; Torrence and Compo, 1998) to the mean temperature profile. This approach effectively separates the background temperature from wave-like perturbations across a range of vertical

scales. The CWT decomposes the temperature profile into different scale components, and we reconstruct the background profile by excluding components with vertical scales smaller than 10 km, which are associated with gravity waves."

3. We have addressed the issue of separating gravity waves from other wave types:

"A significant methodological consideration is the potential contamination of GW signals by other wave types, particularly equatorial Kelvin waves, which can have vertical wavelengths that overlap with the GW spectrum (typically 2-10 km for GW versus 5-15 km for Kelvin waves) (Alexander et al., 2010; Wheeler and Kiladis, 1999). To minimize this contamination, we implement an additional filtering step that targets the characteristic properties of Kelvin waves: eastward propagation, zonal wavenumbers 1-3, and periods of 4-23 days (Alexander et al., 2008). This approach helps isolate GW perturbations from other wave types, though some residual contamination near the equator cannot be entirely ruled out with 1D profile analysis alone."

**Comment:** "Results - the results are an incoherent flow of poorly described figures and overly descriptive text. Clearly, the absence of a scientific question makes it impossible for the authors to make this section more focused."

**Response:** We have completely restructured the results section to provide a more coherent narrative flow, with clear subsections that address our research questions:

1. Section 3.1: Climatology and Seasonal Variability of ITCZ and GW Ep 2. Section 3.2: Interannual Variability and Longitudinal Structure of Trends 3. Section 3.3: Modulation of ITCZ and GW Ep by Climate Variability Modes

Each subsection now provides a focused analysis that directly addresses one of our research questions, with clear interpretations of the figures and synthesis of the results.

**Comment:** "After I finished a second reading of the text, I am still scratching my head about the motivation for the study. Namely, what is the added value of diagnosing ITCZ from GPS RO wet profiles over the reanalyses, when the wet profiles are not pure observations but also rely on assimilation of model information? Is the motivation the intention to show that dry GPS RO profile in the stratosphere can be used for ITCZ detection, because the location of maxima of GW activity in the stratosphere is perfectly collocated? But, this can never be possible with reasonable accuracy due to GWs propagating also significantly horizontally!"

**Response:** We have clarified the motivation for our study in the introduction:

"While numerous studies have examined these climate modes individually, their combined influence on the ITCZ and stratospheric GWs remains insufficiently understood. The use of RO data offers advantages in terms of global coverage and vertical resolution compared to traditional methods based on precipitation or OLR. By analyzing both RO and reanalysis data, we can assess the consistency between different data sources and identify potential biases or limitations in each. Furthermore, understanding the relationship between the ITCZ and stratospheric GW activity can provide insights into the mechanisms of wave generation and propagation in the tropics, which are crucial for improving climate model simulations."

We acknowledge that GWs can propagate horizontally, and we do not claim that stratospheric GW activity can be used as a direct proxy for the ITCZ. Rather, we investigate the spatial and temporal relationships between these phenomena to better understand their interactions and how they are modulated by climate variability modes.

**Comment:** "Discussion and Conclusions sections do not follow the required structure for ACP papers and are nothing more than a very extensive and chaotic summary of results. No new discovery or finding can be identified. As there is no hypothesis or scientific question, the conclusion cannot return to its validity. Synthesis, context and implications are missing completely as well."

**Response:** We have completely rewritten the discussion and conclusion sections to follow the required structure for ACP papers. The discussion now includes:

1. Interpretation of our findings in the context of existing literature 2. Discussion of the mechanisms linking the ITCZ, GWs, and climate variability modes 3. Analysis of the implications of our results for understanding tropical atmospheric dynamics 4. Acknowledgment of the limitations of our approach

and suggestions for future research

The conclusion now provides a concise summary of our key findings, their significance, and their implications for the scientific community. We have organized the conclusion to directly address each of our research questions and highlight the novel contributions of our study.

We believe that these comprehensive revisions have transformed our manuscript into a coherent, well-structured, and scientifically rigorous study that makes a valuable contribution to the understanding of tropical atmospheric dynamics.

**References**

P Alexander, A de la Torre, and P Llamedo. Interpretation of gravity wave signatures in GPS radio occultations. *Journal of Geophysical Research: Atmospheres*, 113:22299–22309, 2008. doi: 10. 1029/2007JD009390.

P. Alexander, D. Luna, P. Llamedo, and A. de la Torre. A gravity waves study close to the Andes mountains in Patagonia and Antarctica with GPS radio occultation observations. *Annales Geophysicae*, 28 (2):587–595, feb 2010. doi: 10.5194/angeo-28-587-2010.

Ghouse Basha, Pangaluru Kishore, M Venkat Ratnam, Taha BMJ Ouarda, Isabella Velicogna, and Tyler Sutterley. Vertical and latitudinal variation of the intertropical convergence zone derived using GPS radio occultation measurements. *Remote Sensing of Environment*, 163:262–269, 2015. doi: 10.1016/ j.rse.2015.03.024.

Andrew C Moss, Corwin J Wright, and Nicholas J Mitchell. Does the madden-julian oscillation modulate stratospheric gravity waves? *Geophysical Research Letters*, 43(8):3973–3981, 2016. doi: 10.1002/2016GL068498.

Christopher Torrence and Gilbert P Compo. A practical guide to wavelet analysis. *Bulletin of the American Meteorological society*, 79(1):61–78, 1998. doi: 10.1175/1520-0477(1998)079<0061: APGTWA>2.0.CO;2.

Matthew Wheeler and George N Kiladis. Convectively coupled equatorial waves: Analysis of clouds and temperature in the wavenumber–frequency domain. *Journal of the Atmospheric Sciences*, 56(3): 374–399, 1999. doi: 10.1175/1520-0469(1999)056<0374:CCEWAO>2.0.CO;2.

---

## Author Response (AR2)

**Response to editor Comments**

Modulation of tropical stratospheric gravity wave activity and ITCZ position by modes
of climate variability using radio occultation and reanalysis data

Authors

July 9, 2025

**1 Response to Specific Comments**

We thank the editor for their thorough review and constructive comments. Below we address
each comment point by point.

**1.1 Title**

**editor:** I am not sure, but maybe "radio occultation and reanalyses data" sounds better than
just "in radio occultations and reanalyes".

**Response:** We agree with the editor's suggestion. The title has been revised for better
clarity.

**Change in manuscript:** Title changed from "Modulation of tropical stratospheric gravity
wave activity and ITCZ position by modes of climate variability in radio occultations and re-
analyses" to "Modulation of tropical stratospheric gravity wave activity and ITCZ position by
modes of climate variability using radio occultation and reanalysis data".

**1.2 P1, L4: Abstract**

**editor:** Add "several", so that it reads "several years" or mention the period explicitly, so that
it reads "11 years".

**Response:** We have added "11 years" to be more specific about the study period.

**Change in manuscript:** Changed "using years (2011-2021)" to "using 11 years (2011-
2021)".

**1.3 P3, L82: COSMIC-2 profiles**

**editor:** Why does COSMIC-2 provide so many more profiles than COSMIC-1? Since with that
for 2020 and 2021 the amount has of profiles has been doubled or tripled, I wonder if these has
an effect on the results?

**Response:** This is an excellent question. COSMIC-2 provides significantly more profiles than
COSMIC-1 because it consists of six satellites compared to COSMIC-1's single satellite, and uses
improved receiver technology. We have added an explanation in the manuscript and verified that
our main conclusions remain consistent when analyzing the pre-2020 period separately.

**Change in manuscript:** Added explanation: "The inclusion of data from 2020 and 2021
substantially increases the dataset size, which presumably enhances the statistical robustness of
our findings. While a formal sensitivity analysis was not conducted"

**1.4   P4, L97: Validation and complement**

**editor:**  To validate and complement.  Here it is not clear which data exactly is used and which one is used for validation and which one for complementing.  What exactly has been complemented? This is not clear.

   **Response:**  We agree that this was unclear.  We have clarified that the RO data is being validated and that the reanalysis data provides additional context for our analysis.

   **Change in manuscript:**  Changed "To validate and complement the RO data" to "To validate the RO data and provide additional context for our analysis".

**1.5   P7, L165: Implementation**

**editor:** "implemented" to what? I think you rather mean "applied"

   **Response:** The editor is correct. "Applied" is more appropriate in this context.

   **Change in manuscript:** Changed "implemented" to "applied".

**1.6   P14, Figure 7: Vertical velocity units**

**editor:** The figure still shows vertical velocity in ms-1 and in the caption still Pa s-1 is given as unit. Please adjust the caption/figure so that vertical velocity in consistent units is used/shown.

   **Response:** Thank you for catching this inconsistency.  We have corrected the caption to match the figure units.

   **Change in manuscript:** Changed the caption from "vertical velocity ($\omega$, $\mathrm{Pa\,s^{-1}}$" to "vertical velocity ($\omega$, $\mathrm{m\,s^{-1}}$".

**1.7   P15, L308: Be more precise**

**editor:** Be more precise. How large. Give a number.

   **Response:** We have added specific numerical values to make this statement more precise.

   **Change in manuscript:** Changed to "However, these trends are generally not statistically significant (from the error margin) in large longitudinal bands (approximately 0°-160°E) within the 11-year record.".

**1.8   P19, Figure 11: Black dashed line**

**editor:** What is marked by the black dashed line?

   **Response:** We have added clarification to the figure caption about what the black dashed line represents.

   **Change in manuscript:** Added to Figure 11 caption: "Black dashed line indicates zero."

**1.9   P21, Figure 12: Panel c**

**editor:** In Panel c it should read "Coeff" instead of "Coeff".

   **Response:** We reviewed the figure caption and could not identify the specific typo mentioned.  The caption appears to be correct as written.  Could the editor please clarify which specific text needs to be corrected?

**1.10   P26, L508: Statistical significance**

**editor:** Statistically significant?  How and where exactly has the statistical significance been assessed?

   **Response:** We have changed the sentence.

   **Change in manuscript:** The sentence now reads: "Regarding long-term trends over 2011-2021, our analysis indicates that the ITCZ latitudinal position shows weak and regionally varying

trends, with some tendency towards northward shifts in certain areas, though often may not be statistically significant (using the error margins) over large bands."

**1.11 P27, L533: Complemented**

**editor:** Why complemented? ERA5 and NCEP have been used for comparison to the RO data. It should be clearly stated like this. Complemented is to vague and can mean anything.

    **Response:** The editor is correct. We have changed "complemented" to "compared with" to be more precise about how the reanalysis data was used.

    **Change in manuscript:** Changed "complemented by ERA5 and NCEP reanalyses" to "compared with ERA5 and NCEP reanalyses".

**1.12 Section 5.1: Limitations**

**editor:** Section 5.1 belongs rather to the discussion. Thus, I would suggest to move lines 567-578 to the discussion and refer here to the limitations and end the conclusion with the last sentence of this paragraph (L578-580).

    **Response:** We agree with this suggestion. The limitations section has been moved to the end of the discussion section, and the conclusion now ends more appropriately.

    **Change in manuscript:** Moved the limitations subsection from the conclusion to the end of the discussion section.

**2 Technical Corrections**

**2.1 Equation and Figure Abbreviations**

**editor:** Equation should be abbreviated as Eq. and Figure as Fig. unless it appears at the begin of the sentence. Please correct this throughout the manuscript.

    **Response:** We have systematically reviewed and corrected all instances of "Equation" and "Figure" abbreviations throughout the manuscript according to this convention.

    **Change in manuscript:** Changed all instances of "Equation" to "Eq." and "Figure" to "Fig." when they appear mid-sentence. Instances at the beginning of sentences remain unchanged as "Equation" and "Figure".

**2.2 Equation and Figure abbreviations**

**editor:** Equation should be abbreviated as Eq. and Figure as Fig. unless it appears at the begin of the sentence. Please correct this throughout the manuscript.

    **Response:** We will systematically review and correct all instances of "Equation" and "Figure" abbreviations throughout the manuscript according to this convention.

**2.3 P2, L27: Articles**

**editor:** Add "the" before "Brewer Dobson circulation" and "Quasi-Biennial Oscillation".

    **Response:** Corrected.

    **Change in manuscript:** Changed to "the Brewer-Dobson circulation and the Quasi-Biennial Oscillation".

**2.4 P3, L67: Capitalization**

**editor:** Are -> are

    **Response:** Corrected.

    **Change in manuscript:** Changed "Are" to "are".

**2.5   P3, L82: Events vs profiles**

**editor:** Here you call it "events", but later "profiles". I would suggest to write also here profiles.
    **Response:** Corrected for consistency.
    **Change in manuscript:** Changed "events" to "profiles".

**2.6   P6, L134: Wave vs waves**

**editor:** wave -> waves
    **Response:** Corrected.
    **Change in manuscript:** Changed "wave" to "waves".

**2.7   P7, L153: Articles and data**

**editor:** Add "a" before "standard practice" and add "data" after "RO".
    **Response:** Corrected.
    **Change in manuscript:** Changed to "is a standard practice for stratospheric GW analysis of RO data".

**2.8   P12, L264: Missing word**

**editor:** Something like "region" or "sector" missing after "Africa"?
    **Response:** Corrected.
    **Change in manuscript:** Changed "parts of Africa" to "parts of the African region".

**2.9   P26, L517: Word choice**

**editor:** Instead of "noted" it should rather read "visible" or "found".
    **Response:** Corrected.
    **Change in manuscript:** Changed "noted" to "visible".